# Policy Learning from Tutorial Books
# via Understanding, Rehearsing and Introspecting

Xiong-Hui Chen[1,+,*] Ziyan Wang[2,*], Yali Du[2,◇], Shengyi Jiang[5],
Meng Fang[4], Yang Yu[1,◇], Jun Wang[3,◇]

[1] National Key Laboratory for Novel Software Technology, Nanjing University, China & School of Artificial Intelligence, Nanjing University, China [2] Cooperative AI Lab, Department of Informatics, King's College London [3] AI Centre, Department of Computer Science, University College London [4] University of Liverpool [5] The University of Hong Kong

## Abstract

When humans need to learn a new skill, we can acquire knowledge through written books, including textbooks, tutorials, etc. However, current research for decision-making, like reinforcement learning (RL), has primarily required numerous real interactions with the target environment to learn a skill, while failing to utilize the existing knowledge already summarized in the text. The success of Large Language Models (LLMs) sheds light on utilizing such knowledge behind the books. In this paper, we discuss a new policy learning problem called **P**olicy **L**earning from tutorial **B**ooks (PLfB) upon the shoulders of LLMs' systems, which aims to leverage rich resources such as tutorial books to derive a policy network. Inspired by how humans learn from books, we solve the problem via a three-stage framework: **U**nderstanding, **R**ehearsing, and **I**ntrospecting (URI). In particular, it first rehearses decision-making trajectories based on the derived knowledge after understanding the books, then introspects about the imaginary dataset to distill a policy network. We build two benchmarks for PLfB based on Tic-Tac-Toe and Football games. In the experiment, URI's policy achieves a minimum of 44% net winning rate against GPT-based agents without any real data. In the much more complex football game, URI's policy beat the built-in AIs with a 37% winning rate while GPT-based agents can only achieve a 6% winning rate. The project page: plfb-football.github.io.

## 1 Introduction

Humans can acquire new skills through various written materials that provide condensed knowledge, without the need for direct interactions with the target environment. In contrast, traditional policy learning paradigms, such as reinforcement learning (RL) [1] primarily rely on trial and error [2–4]. Despite recent advances in offline RL [5, 6] showing that policy improvements can be achieved simply by using pre-collected data, in fields such as embodied AI for robotics [7], the process of collecting large amounts of decision-making data remains costly and, at times, impractical or even impossible. Therefore, a question arises: *similar to how humans learn, can a policy learn from non-real data, e.g., from tutorial books?*

We argue that the recent successes of Large Language Models (LLMs), such as GPT-4 [8], and LLaMA [9] already demonstrated the potential to learn from textual content. However, current studies focus on using LLM directly for decision making [10, 11] or integrating them as auxiliary modules in other machine learning workflows [12–14]. In this study, we introduce a novel topic built upon the shoulders of LLMs' systems: Policy Learning from Books (PLfB). PLfB aims to derive a policy

---

* Equal Contribution. + Work done during Xiong-Hui Chen's visit at King's College London.
◇ Corresponding: yali.du@kcl.ac.uk, yuy@nju.edu.cn, jun.wang@cs.ucl.ac.uk.

38th Conference on Neural Information Processing Systems (NeurIPS 2024).

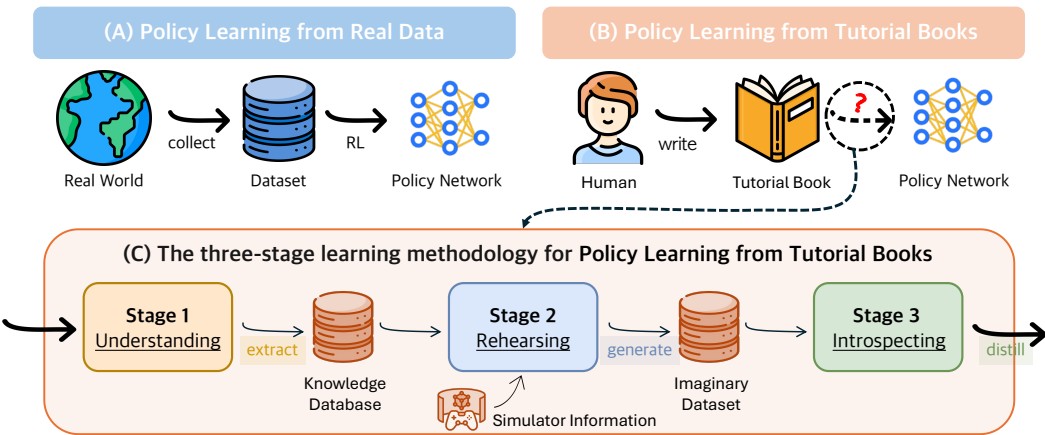

**Figure 1:** (A-B) A comparison between the problem of policy learning from tutorial books (PLfB) and from data; (C) An illustration of the understanding, rehearsal, and introspection for PLfB.

network directly from natural language texts, bypassing the need for numerous real-world interaction data, as shown in Fig. 1(A-B). *This can be viewed as a further step towards enabling more resources for policy learning and also a more generalized form of offline RL problem, which uses textbooks to learn a policy offline.* The essential challenge of PLfB comes from the inevitable large modality gaps between the text space, which includes the knowledge related to decision-making, and the policy network space, which formulates the parameters of the policy function.

To realize PLfB, inspired by human learning processes, we propose a three-stage learning methodology: understanding, rehearsal, and introspection (URI), which is shown in Fig. 1(C). For understanding, it first extracts knowledge from books to form a knowledge database; then it rehearses imaginary decision-making trajectories with the help of the knowledge retrieved from the database; finally, it introspects on the imaginary dataset to distill a policy network for decision-making. We present the first practical implementation of URI, employing LLMs to convert book paragraphs into pseudocode for policy, dynamics, and reward functions. This pseudocode forms a code-based knowledge database, which is used to generate an imaginary dataset based on retrieval augmented generation techniques [15]. A policy learning technique inspired by offline RL [6] is then applied to distill a policy network that addresses inaccuracies in the imagined actions, rewards, and states.

In the experiments, we build two benchmarks from Tic-Tac-Toe and Football games for PLfB. We first validate URI on Tic-Tac-Toe game, where the tutorials are manually constructed to cover the complete knowledge for decision-makings. The results show that our method achieves at least 44% net win rate against GPT-based agents without any real game interaction. In addition, we build a testbed based on football tasks, which is popular in recent studies [16] as a difficult decision-making task, and focus on policy learning from football tutorial books, which naturally contain condensed knowledge, especially information closely related to football skill acquisition, the environment and dynamics of football games, and ways to evaluate football behaviors. We collect football tutorial books from RedPajama [17]. The policy is deployed in the Google Football simulator [18] directly. Our agent controlled by the policy network could beat the built-in AI with a 37% winning rate on average while using GPT as the football agent can only achieve a 6% winning rate. The experiments demonstrate the scalability of our approach from simple board games to complex scenarios.

## 2 Related Work

LLMs have demonstrated remarkable potential in achieving human-level ability in various tasks, sparking a surge in studies investigating LLM-based autonomous agents. Based on the different roles of LLM in the agents, there are two types of work. Firstly, LLM can be used as an actionable agent directly [10]. Early works [19–21] prefer open-loop plans and hard-code search algorithms to improve the long-term planning ability of LLM. Closed-loop planning [22–24] leverages environmental feedback, thereby facilitating more adaptive decision-making. Secondly, LLMs can be used to assist or accelerate the learning of agents. The form of assistance can be quite diverse. It can generate high-level plans [25, 26], (surrogate) rewards [27–32], transitions [33–35], or as an adapter to convert human instructions into structured inputs [36]. In our work, LLMs play a different role compared to all the above-mentioned types. LLMs are to understand the knowledge in the books, rehearsing the decision-making process to derive an imaginary dataset based on the knowledge. The policy to control the agent is a simple neural network distilled from the imaginary data.

# 3 Preliminaries

**Markov Decision Process (MDP)** is defined as a tuple of $\mathcal{M} := (\mathcal{S}, \mathcal{A}, T, R, \gamma, \rho_0)$ [1] of the target environment, where $\mathcal{S}$ is state space, $\mathcal{A}$ is action space, $T : \mathcal{S} \times \mathcal{A} \to \Delta(\mathcal{S})$ is the transition function ($\Delta(\cdot)$ is the probability simplex), $R : \mathcal{S} \times \mathcal{A} \to [0, R_{\max}]$ is the reward function, $\gamma \in [0, 1)$ is the discount factor, and $\rho_0$ is the initial distribution over states. A policy $\pi : \mathcal{S} \to \Delta(\mathcal{A})$ describes the distribution of actions for each state.

**Retrieval Augmented Generation (RAG)** [15] has shown great ability to improve LLM's generation ability for knowledge-intensive tasks. RAG enables LLMs to query external data sources to obtain relevant information before proceeding to answer questions or generate text. Formally, given an LLM $\mathbf{M}$, the retrieval module $\mathbf{R}(x, \{y_i\})$ takes a textual query $x$ as input, finds the most relevant textual segment $y^*$ by finding the most similar $\mathbf{E}(y)$ from the knowledge database $y \in \{y_i\}$ compared to $\mathbf{E}(x)$. Augmentation transforms the content into an additional input for the LLM's generation. In this paper, RAG is used to implement the URI algorithm. We use standard RAG techniques as the retrieval module, i.e., cosine similarity matching [37, 38], within the implementation of URI. In particular, we use GPT embedding to index the texts in the database and also the current query, then use cosine similarity to find the top-$n$ matching data for downstream tasks, i.e., $\mathbf{M}(x, [y_1^*, ..., y_n^*])$.

**Offline RL** [39] addresses the problem of learning policies from a pre-collected dataset $\mathcal{D}$. Existing studies can be classified into two categories: model-free and model-based methods. Model-free [5, 40–45] methods learn a policy directly from the dataset $\mathcal{D}$ through a specially designed conservative policy learning loss which usually aims to avoid policy taking actions unseen in $\mathcal{D}$. Model-based offline algorithms [6, 46–52] first estimate the dynamics and reward model $\hat{T}$ and $\hat{R}$ from the dataset $\mathcal{D}$. The policy is learned by iterating with $\hat{T}$ and $\hat{R}$. In the process, specially designed penalties $\mathcal{R}$ are adopted to discourage the policy from visiting states where the model predictions are of high uncertainty.

# 4 Problem Formulation of Policy Learning from Tutorial Books

The goal of Policy Learning from Tutorial Books (PLfB) is to use an algorithm ALG to learn a policy $\hat{\pi}^* = \text{ALG}(\mathcal{B}, |\mathcal{M}|)$ from textual books $\mathcal{B}$, which can maximize the cumulative discounted reward $\eta(\pi) = \sum_t \mathbb{E}_{a_t \sim \pi} \gamma^t R(s_t, a_t)$, where the book is defined as $N_b$ segments $b_i$ divided by paragraphs, i.e., $\mathcal{B} := \{b_1, ..., b_i, ..., b_{N_b}\}$, and $|\mathcal{M}|$ denotes the brief textual descriptions of the structures of MDP, including the descriptions of state space, action space, the task we faced, and the initial state distribution. $|\mathcal{M}|$ is inevitable to be used since $\mathcal{B}$ only gives general knowledge of the target environment while $|\mathcal{M}|$ defines the specific information, e.g., the exact format to interact with the simulator. Unlike standard RL, during the learning process of PLfB, the algorithm has no access to the environment. To ensure the feasibility of extracting a non-trivial policy, we make a mild assumption that $\mathcal{B}$ contains the descriptions of transition functions $\{\hat{T}\}$, and reward functions $\{\hat{R}\}$ that can be regarded as the approximations of $T$ and $R$, respectively. These descriptions could range from simple game rules to complex natural language paragraphs, depending on the books. It also contains descriptions of relatively high-quality behavior policies.

# 5 Understanding, Rehearsing, and Introspecting

In this section, we first present our motivation for the proposed three-stage framework in Sec. 5.1. After that, we explain how we implement those stages in Sec. 5.2, 5.3, and 5.4 respectively.

## 5.1 Motivation of the Three-Stage Framework

As shown in Fig. 1, the methodology to solve PLfB consists of three major stages: understanding, rehearsing, and introspecting (URI). The motivation of these three modules can be easily understood by reviewing how humans learn from books: as humans, we first extract knowledge from books to extend our knowledge database. Then, to acquire skills, we often rehearse the possible consequences of applying the skill in our mind, integrating knowledge from the books with our prior life knowledge. Finally, we will re-examine the steps we took in our minds and think how we could have done better until we confirm that we know how to execute in the real world. In this paper, we mimic the above three steps via the following modules: **Understanding** module takes paragraphs of books as input and forms a knowledge database organized in pseudo-code. **Rehearsing** module iteratively takes the current imagined state as input and outputs the action, next state, and reward with guidance from the

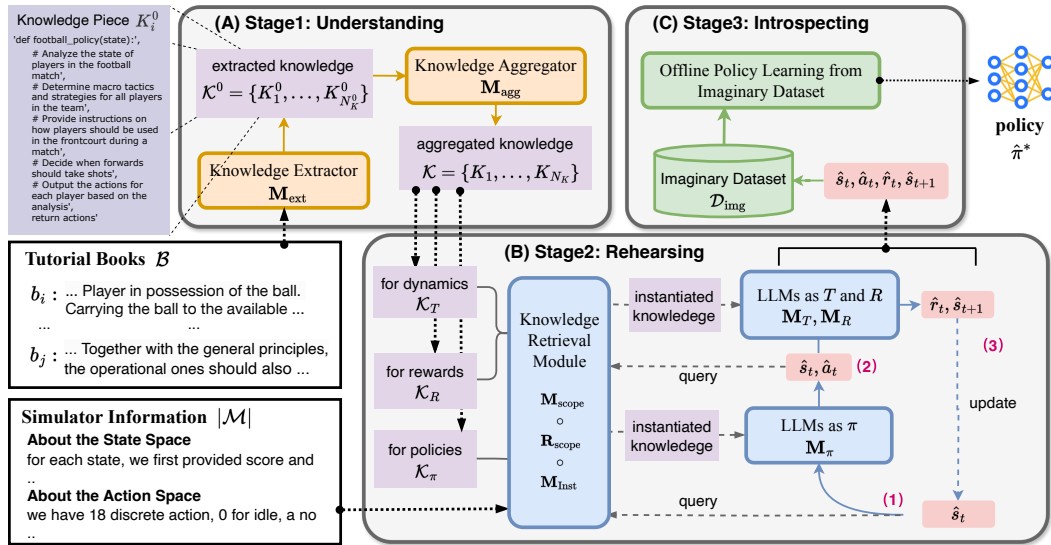

**Figure 2:** The URI pipeline consists of three major stages: (A) **Understanding:** The knowledge extractor and aggregator modules process paragraphs from books to form a structured knowledge database organized as pseudo-code. (B) **Rehearsing:** Using the knowledge database, the simulator generates and iterates through imagined states, actions, and rewards to create an extensive imaginary dataset. (C) **Introspecting:** The introspection module refines the policy network by evaluating and correcting errors in the generated states, actions, and rewards to ensure accurate and effective policy implementation. The pseudocode of the pipeline is in Appendix F.8.

database. After gathering this imagined content to form a dataset, **Introspecting** module distills a policy network, which should consider errors of generations of state, action, and rewards.

We give the overall architecture of our implementation of URI in Fig. 2. A detailed description of how they realize these functionalities will be discussed in the following.

## 5.2 Book Content Understanding

The understanding module is responsible for extracting knowledge $\mathcal{K} := \{K_1, ..., K_i, ..., K_{N_K}\}$ from the books $\mathcal{B}$, where $K_i$ denotes one piece of knowledge. For we humans, the knowledge is naturally stored in our brains. For machines, the first question is what should be the appropriate format to represent the knowledge. Since LLM has shown superior performance in code generation and codes themselves as interpretable, compact, and expressive languages [53], we also choose it as the basic format of knowledge representation. Different from previous works [10] which directly asked for runnable code for the downstream tasks using, we do not execute this code. Instead, we just use it as a more flexible and abstract description of knowledge which still maintains a rigorous control flow. As shown in Fig. 2(A), a knowledge extractor module is used as the first step. The knowledge extractor is an LLM-injected model. We iteratively ask $K_j = \mathbf{M}_{\text{ext}}(b_i)$ to examine whether a paragraph is related to the decision-making elements, i.e., policy functions, reward functions, dynamics functions, and how to represent them as pseudocode.

Moreover, humans often learn by repeatedly reading the texts across pages and even books, updating existing knowledge with newly learned ones, and summarizing them into more general and abstract forms. A similar procedure is achieved by the knowledge aggregator module, shown in Fig. 2(A), as the second step of understanding. The knowledge aggregator is also an LLM-injected model. We iteratively ask $\mathbf{M}_{\text{agg}}([K_i, K_j, K_l, ...])$ to aggregate relevant knowledge among different segments based on a similarity estimation of the embedding model $\mathbf{E}$. Formally, let $\mathcal{K}^0 := \{K_1^0, ..., K_{N_K^0}^0\}$ be the paragraph-wise knowledge extracted by $\mathbf{M}_{\text{ext}}$. For the $j$-the iteration, $[K_o^{j+1}, K_p^{j+1}, ...] = \mathbf{M}_{\text{agg}}([K_i^j, K_l^j, K_n^j, ...])$, where $[K_i^j, K_l^j, K_m^j, ...]$ is from $\mathcal{K}^j$ by selecting the most similar $N_{\text{agg}}$ pieces of knowledge by comparing their cosine similarity under the embedding model $\mathbf{E}(K)$ of GPT and $o, p, i, l, n$ here denote the indexes. $[K_o^{j+1}, K_p^{j+1}, ...]$ are then added to $\mathcal{K}^{j+1}$. The iteration will stop if $|\mathcal{K}^{j+1}| >= |\mathcal{K}^j|$, where $|\mathcal{K}^j|$ is the pieces of knowledge at $j$-th iteration. After iteration stops, the remaining knowledge pieces constitute the knowledge databases by splitting them into dynamics-related knowledge $\mathcal{K}_T$, reward-related knowledge $\mathcal{K}_R$, and policy-related knowledge $\mathcal{K}_\pi$ for later modules to use. More details are provided in Appendix F.1.

## 5.3 Knowledge-based Rehearsing of Decision-Making

When humans develop a rough understanding of a new skill $\pi$ from knowledge $\mathcal{K}$, they will usually imagine what choice they will make given certain situations and what are the possible consequences in their minds. This rehearsing procedure would help humans correct apparent mistakes and consider better actions for long-term benefits [54, 55].

We implement a closed-loop generation process involving the LLM-injected dynamics function $\mathbf{M}_T$, reward function $\mathbf{M}_R$, and policy $\mathbf{M}_\pi$, as depicted in Fig. 2(B). This approach resembles the model rollout in traditional model-based RL, where the policy, reward function, and dynamics function are represented by LLMs. Given the current imagined state $\hat{s}_t$, the LLMs $\mathbf{M}_\pi$ first predict the most plausible action $\hat{a}_t$. Subsequently, $\mathbf{M}_T$ and $\mathbf{M}_R$ generate the next state $\hat{s}_{t+1}$ and the associated reward $\hat{r}_t$ based on this action and the current state. In the process, a knowledge retrieval module is involved in selecting relevant knowledge pieces enhancing the LLM input with this information for LLM's predictions, e.g., $\hat{a}_t = \mathbf{M}_\pi(\hat{s}_t, \mathbf{R}(s_t, \mathcal{K}_\pi))$, $\hat{r}_t = \mathbf{M}_R(\hat{s}_t, \hat{a}_t, \mathbf{R}(s_t, \mathcal{K}_R))$ and $\hat{s}_{t+1} = \mathbf{M}_T(\hat{s}_t, \hat{a}_t, \mathbf{R}(s_t, \mathcal{K}_T))$. The knowledge retrieval module includes the following two steps:

**State-based Knowledge Scope Retrieval:** The fundamental problem of standard RAG techniques, i.e., embedding-vector similarity matching, in this scenario, is the modality gap between the query $s_t$ and the knowledge $\mathcal{K}$. Standard RAG approaches aim to identify information closely related to the query, while here we need to ask the most suitable knowledge to be applied as a dynamics/reward/policy function *that has the best predictions* for the queried state and action. Standard RAG techniques tend to retrieve knowledge pieces that include similar text patterns as the queried states and fail to find the best knowledge for predictions. To address this, we propose a knowledge scope retrieval method that includes a simple yet effective preprocessing step to bridge the modality gap. In particular, we traverse all knowledge pieces $K_i \in \mathcal{K}$ by iteratively sampling $n_{\text{scope}}$ knowledge pieces $\{K_i\}_{n_{\text{scope}}}$ from the database $\mathcal{K}$, combining them with simulator information $|\mathcal{M}|$ and using LLM $\{K_i^S\}_{n_{\text{scope}}} = \mathbf{M}_{\text{scope}}(\{K_i\}_{n_{\text{scope}}}, |\mathcal{M}|)$ to determine the preferred scopes $K_i^S$ for each piece of knowledge. $K_i^S$ is defined by the preferred subspace of state to use the knowledge. Then a standard RAG technique is applied to identify the most relevant knowledge scope $K_j^s$ and its relevant knowledge $K_j$. Formally, $(\{K_j\}, \{K_j^S\}) = \mathbf{R}_{\text{scope}}(\hat{s}, \mathcal{K}^S)$, where $\mathcal{K}^S$ is the scopes of the knowledge database. This method is effective for embedding models in retrieving the correct knowledge as the texts of keys and queries both are about the descriptions of states.

**Post-Retrieval: Knowledge Instantiation:** Predicting based on code knowledge requires the LLM's robust understanding of code. We refine this process employing an LLM to instantiate the code $K^I = \mathbf{M}_{\text{Inst}}(\hat{s}, |\mathcal{M}|, \{K_j\})$ based on the current state $\hat{s}$, the simulator information description $|\mathcal{M}|$ and the knowledge $\{K_j\}$ retrieved by $\mathbf{R}_{\text{scope}}$. Knowledge instantiation involves generating a domain-specific pseudocode based on the knowledge coded in the retrieved information, tailored to the target environment's current state and action requirements.

Finally, three LLMs are involved to generate the imaginary dataset $\mathcal{D}_{\text{img}}$, including $\hat{a}_t = \mathbf{M}_\pi(\hat{s}_t, \mathbf{M}_{\text{Inst}}(s_t, |\mathcal{M}|, \mathbf{R}_{\text{scope}}(\hat{s}, \mathcal{K}_\pi^S)))$, $\hat{r}_t = \mathbf{M}_R(\hat{s}_t, \hat{a}_t, \mathbf{M}_{\text{Inst}}(s_t, |\mathcal{M}|, \mathbf{R}_{\text{scope}}(\hat{s}, \mathcal{K}_R^S)))$ and $\hat{s}_{t+1} = \mathbf{M}_T(\hat{s}_t, \hat{a}_t, \mathbf{M}_{\text{Inst}}(s_t, |\mathcal{M}|, \mathbf{R}_{\text{scope}}(\hat{s}, \mathcal{K}_T^S)))$, where $\mathcal{K}_\pi^S$, $\mathcal{K}_R^S$, and $\mathcal{K}_T^S$ are the scope of the knowledge database $\mathcal{K}_\pi$, $\mathcal{K}_R$, and $\mathcal{K}_T$ respectively. To start the rollout of a trajectory, we need a state as the initial state. It has several choices to achieve, e.g., sampled from $\mathcal{S}$, generated by another LLM, or pre-collected a small number of real states from the target environments as part of simulator information, i.e., initial state distribution $\rho_0$. Since the first choice might introduce unrealistic initial states in complex scenarios, in Football tasks, we opt for a more practical approach by pre-collecting a small set of real states from the target environments for our experimental validation. The detailed setup is in the experiment section. Besides, during the rollout over $H$ steps, we reuse instantiated knowledge to reduce the overload of LLM's callings. More details are provided in Appendix F.2.

## 5.4 Introspecting based on the Imaginary Dataset

The direct output LLM in the rehearsing stage can be sub-optimal or incorrect. We would like to re-examine the collected data and try to distill a policy that can avoid the side effects. We can regard the data $\mathcal{D}_{\text{img}}$ collected during the rehearsing stage as an offline dataset and apply offline RL algorithms to train an improved policy from the dataset. However, directly applying existing offline RL algorithms over-simplifies the problem. Compared with the standard offline setting where only the behavior policy is sub-optimal, there is an additional misalignment in the data generated

during the rehearsal: the transition and the reward function estimated by the LLM are also inaccurate. Overlooking such inaccuracy would result in a policy exploiting the sub-optimal transition and reward function and cause performance degradation or even risky behaviors in the final deployment.

To solve the problem, in this paper, we adopt the Conservative Q-learning [5] as the base offline RL algorithm, whose learning objective is as follows:

$$\min_Q \max_\pi \; \alpha \left( \mathbb{E}_{\hat{s} \sim \mathcal{D}_{\text{img}}, a \sim \pi(a|\hat{s})} [Q(\hat{s}, a)] - \mathbb{E}_{\hat{s}, \hat{a} \sim \mathcal{D}_{\text{img}}} [Q(\hat{s}, \hat{a})] + \mathcal{R}(\pi) \right)$$
$$+ \mathbb{E}_{\hat{s}, \hat{a} \sim \mathcal{D}_{\text{img}}} [(Q(\hat{s}, \hat{a}) - \hat{\mathcal{B}}^\pi \hat{Q}(\hat{s}, \hat{a}))^2],$$

where $\hat{\mathcal{B}}^\pi \hat{Q}$ is the bellman update operator to update the Q-value function [1], and the first term is to learn a policy with conservative Q value [5]. As a solution of introspecting from imaginary data, as shown in Fig. 2(C), we add the uncertainty of the reward and transition estimation as the regularization terms, $\mathcal{R}_R$ and $\mathcal{R}_T$ over the original reward $\hat{r}$ output by the LLM $\mathbf{M}_R$. In practice, we adopt these regularization terms by applying them when we backup the $\hat{Q}^k$:

$$\hat{\mathcal{B}}_I^\pi \hat{Q}(\hat{s}, \hat{a}) := \hat{r} - \eta_R \mathcal{R}_R(\hat{s}, \hat{a}) - \eta_T \mathcal{R}_T(\hat{s}, \hat{a}) + \gamma \mathbb{E}_{\hat{s}' \sim \mathcal{D}_{\text{img}}, a' \sim \pi_k(a'|\hat{s}')} [Q(\hat{s}', a')],$$

where $\hat{s}' \sim \mathcal{D}_{\text{img}}$ is to sample the next state given $\hat{s}, \hat{a}$, $\eta_R$ and $\eta_T$ are two hyper-parameters to control the weighting of the uncertainty terms. Inspired by Model-based Offline Policy Optimization (MOPO) [6], the uncertainty is estimated by an ensemble of $N_{\text{ens}}$ Gaussian models of $\hat{T}$ and $\hat{R}$, which is learned by maximizing logarithmic likelihood from the imaginary dataset. Then the uncertainty is estimated by $\mathcal{R}_R(s, a) = \max_{i \in \{1, ..., N_{\text{ens}}\}} \sigma_i^r(s, a)$ and $\mathcal{R}_T(s, a) = \max_{i \in \{1, ..., N_{\text{ens}}\}} \sigma_i^T(s, a)$, where $\sigma_i^r$ and $\sigma_i^T$ are the $i$-th reward and dynamics model's predicted standard deviation for $s, a$ respectively. We call the solution of CQL with $\hat{\mathcal{B}}_I^\pi$ as the bellman update operator Conservative Imaginary Q-Learning (CIQL). This regularization is easy to implement and can force the policy to generalize better over the regions where the LLM outputs inconsistent next states and rewards.

## 6 Experiments

We build two benchmarks with increasing complexity to evaluate PLfB: a classical Tic-Tac-Toe game with discrete states and deterministic rules, and the Google Research Football environment (GRF) [56] featuring continuous states and multi-agent interactions. In Sec. 6.1, we introduce the experimental settings for both environments. In Sec. 6.2, we analyze the performance of URI against various baselines. In Sec. 6.3 and 6.4, we examine the effectiveness of our knowledge extraction and retrieval mechanisms. Finally, in Sec. 6.5-6.8, we provide a detailed analysis of the training process, including data generation, ablation studies, inference efficiency, and dataset visualization.

### 6.1 Experiment Setups

**Tic-Tac-Toe Game**    To build a proof-of-concept benchmark for PLfB, we first construct a Tic-Tac-Toe (TTT) environment where players take turns placing their marks (X or O) on a 3×3 grid, aiming to form a line of three marks. This classical game serves as an ideal testbed for two reasons: (1) it has a known optimal solution (minimax algorithm), allowing us to evaluate how close our learned policy is to optimality; (2) due to its discrete and deterministic nature, we can collect a complete set of optimal trajectories covering all possible game states. Based on these trajectories, we can use GPT to generate comprehensive tutorial texts that contain complete knowledge of game mechanics and winning strategies, thereby controlling for the impact of knowledge incompleteness on algorithm performance. In our experiments, each policy plays as 'X' and moves first, with performance measured by win rate, draw rate, loss rate, and net win rate (win rate minus loss rate) across different opponents. For detailed game rules and setup, please refer to Appendix A.

**Google Research Football [56]**    is a physics-based 3D football simulator that supports the main football rules such as goals, fouls, corners, penalty kicks, and offside. Google Research Football (GRF) includes a built-in AI bot for the opposing team, whose difficulty can be adjusted between 0 and 1. An illustration of the game can be seen in Fig. 7. We define three custom difficulty levels for our experiments on the 11vs11 scenario: easy, medium, and hard. The difficulty levels differ in the bot's reaction time and decision-making defined in GRF, with higher difficulty corresponding to a stronger opponent. The major metric in our experiment is **Goal Difference per Match** (GDM), calculated as the average number of goals scored in all league matches minus the average number of

**Table 1:** Performance of different policies in Tic-Tac-Toe. Each policy plays as 'X' and moves first, tested across 100 matches (50 for LLM-based methods).

| Player X | LLM-as-agent | | | | LLM-RAG | | | | Random Policy | | | | Minimax-noise | | | |
|---|---|---|---|---|---|---|---|---|---|---|---|---|---|---|---|---|
| | W | D | L | W-L | W | D | L | W-L | W | D | L | W-L | W | D | L | W-L |
| *LLM-as-agent* | - | - | - | - | 58% | 22% | 20% | +38% | 52% | 6% | 42% | +10% | 16% | 4% | 80% | -64% |
| *LLM-RAG* | 72% | 18% | 10% | +62% | - | - | - | - | 52% | 14% | 34% | +18% | 10% | 18% | 72% | -62% |
| **URI (Ours)** | **80%** | 6% | 14% | **+66%** | **62%** | 10% | 18% | **+44%** | **70%** | 12% | 18% | **+52%** | **36%** | 54% | 10% | **+26%** |
| **Minimax (Optimal)** | 84% | 16% | 0% | +84% | 68% | 32% | 0% | +68% | 78% | 22% | 0% | +78% | 34% | 66% | 0% | +34% |

goals conceded per match. For URI, for each seed, we selected the average performance of the top 3 checkpoints among all recorded checkpoints as the final performance. For more details about the GRF implementation, please refer to Appendix B.

**Datasets** We utilize three datasets in our approach. For Tic-Tac-Toe, we generate tutorial texts by having GPT analyze all possible game trajectories collected from an optimal minimax policy (see Appendix A.2). For football, we collect the textbook dataset from the open-source book dataset RedPajama-1T [57], focusing on titles and abstracts related to football or soccer. After filtering, we obtain a curated set of ninety books closely aligned with the domain. For both environments, we need initial states to start our imaginary data generation process. In Tic-Tac-Toe, since we play as 'X' and move first, we always start from an empty board. In GRF, we sample 7,500 states from the rollout of a rule-based policy competing with the hard built-in AI. Due to limited resources, to distill the policy, we imagine 75,000 transitions via URI, which is 10 times compared to the initial states.

**Baselines** We compare URI against several baselines across both environments. **LLM-as-agent** directly uses a large language model (GPT 3.5) to output actions conditioned on the current state description. **LLM-RAG** enhances LLM-as-agent by retrieving relevant knowledge from the database extracted from the tutorial books, similar to the retrieval step in our rehearsing stage, but directly outputs the action without policy learning. For Tic-Tac-Toe, we additionally include **Minimax-noise**, which follows the optimal minimax strategy but with 30% random action selection to serve as a near-optimal baseline. For GRF, we include **Rule-based-AI** from the Kaggle Football Competition [58], which is hand-designed and serves as a reference for the performance of hand-coded policies. **Random Policy** randomly chooses actions in both environments. For the implementation details about the baselines, please refer to Appendix D.

## 6.2 Policy Performance Evaluation

**TTT Results** We first evaluate URI's performance in the Tic-Tac-Toe environment. Tab. 1 shows the head-to-head match results between different policies, where each policy plays as 'X' and goes first. URI demonstrates superior performance across all opponents, achieving net win rates (win rate minus loss rate) of +66%, +44%, and +52% against LLM-as-agent, LLM-RAG, and Random Policy respectively. Moreover, when playing against Minimax-noise, which introduces 30% randomness to the optimal minimax strategy, URI maintains a positive net win rate of +26%, while all other baselines suffer negative net win rates. This indicates URI's ability to learn effective strategies from tutorial texts even in this classical game setting.

**GRF Results** The results in Tab. 2 demonstrate the superiority of the proposed URI approach compared to the baselines in the 11 vs 11 full-game scenarios of the GRF environment. The LLM-based agents, including LLM-as-agent and LLM-RAG, exhibit zero-shot task completion capabilities, outperforming the Random Policy. However, even with the use of RAG techniques, the best-performing LLM agent can only barely match the performance of the Medium-level built-in AI and fails to achieve any wins against the Hard-level built-in AI. In contrast, URI surpasses the performance of the baseline methods on all difficulty levels. Surprisingly, in the Hard task, URI achieves a higher win rate than the Rule-based Policy. We believe this is due to URI's ability to leverage knowledge from domain textbooks and generate high-quality imaginary data for policy learning, enabling it to learn more adaptable and robust policies compared to the hand-crafted rule-based AI.

These results across both simple and complex environments highlight the scalability of URI in learning strong policies from well-defined board games to challenging scenarios.

## 6.3 Effectiveness of Code-based Knowledge Extraction and Aggregation

**Table 2:** Performance Comparison of Different Policies Against Built-in AI Levels in a GRF 11 vs 11 settings, where the performance of URI is averaged among three different seeds, LLM-as-agent, LLM-RAG is tested with 10 matches, and URI policy and random policy is tested with 40 matches.

| Level | | LLM-as-agent | LLM-RAG | Random Policy | URI (Ours) | Rule-based-AI |
|---|---|---|---|---|---|---|
| Easy | Win | 20% | 30% | 2% | **37% ± 4%** | 70% |
| | Draw | 60% | 60% | 55% | 57% ± 4% | 30% |
| | Lose | 20% | 10% | 43% | 6% ± 4% | 0% |
| | GDM | 0.0 | 0.2 | -0.58 | **0.40 ± 0.14** | 0.7 |
| Medium | Win | 0% | 20% | 2% | **42% ± 12%** | 70% |
| | Draw | 60% | 60% | 43% | 50% ± 8% | 30% |
| | Lose | 40% | 20% | 55% | 8% ± 4% | 0% |
| | GDM | -0.4 | 0.0 | -0.76 | **0.43 ± 0.24** | 0.7 |
| Hard | Win | 0% | 0% | 3% | **32% ± 14%** | 30% |
| | Draw | 50% | 40% | 43% | 58% ± 6% | 70% |
| | Lose | 50% | 60% | 53% | 10% ± 7% | 0% |
| | GDM | -0.5 | -0.6 | -0.73 | **0.32 ± 0.14** | 0.3 |
| Average | Win | 6.7% ± 9.4% | 16.7% ± 12.5% | 2.3% ± 0.5% | **40.3% ± 6.2%** | 56% |
| | GDM | -0.30 ± 0.22 | -0.13 ± 0.34 | -0.69 ± 0.08 | **0.38 ± 0.05** | 0.56 |

According to Sec. 5.2, we perform an iterative process of code extraction and aggregation to understand the tutorial books and obtain executable knowledge. As shown in Fig. 3(a), for Tic-Tac-Toe, the initial extraction yields approximately 600 code segments which are effectively consolidated through the aggregation process. Similarly, in the more complex football domain (Fig. 3(b)), the number of code segments for the dynamics, policy, and reward functions decreases significantly over the aggregation

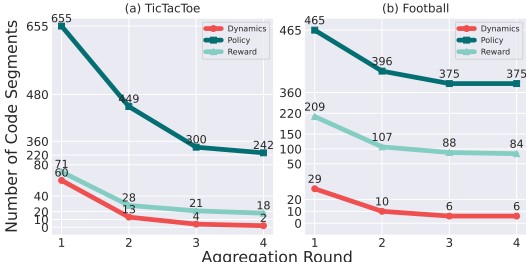

**Figure 3:** Knowledge Segment Aggregation.

rounds. Through iterative aggregation, the number of code segments, decreases significantly in both environments, which helps condense the extracted knowledge into a more compact and coherent form. This consistent pattern across domains of varying complexity demonstrates the robustness of our knowledge extraction and aggregation approach.

### 6.4 Correlation between Knowledge Embedding and Current State Embedding

To validate the effectiveness of our code embedding method mentioned in Sec. 5.3, we conducted experiments in GRF task comparing it with two baselines: **Vector Embedding**, which directly compares the state and code embeddings, and **Code Summary**, which first summarizes the code segments before comparing the embeddings. We evaluate the top-15 hit rate in a hand-crafted test dataset. The observations and actions in the dataset we used are collected by the rule-based policy interacting with the real environment, while the ground-truth codes are labeled by GPT and aggregated using a similar way as in Sec. 5.2. As shown in Fig. 4(a), our method significantly outperforms the other two baselines on both pre-trained language models. The hit rate, which measures the proportion of relevant code segments retrieved, is consistently higher for URI across all three random seeds, demonstrating its robustness and superiority in improving the correlation between the state embedding and the code embedding. These results highlight the importance of learning a dedicated matcher for effective code retrieval in our framework.

### 6.5 Tracing the Data Generation Process

In URI, it is crucial to guarantee that the imaginary data include states, actions, and rewards. The task is non-trivial since it has to involve several transformations to align the gap between textual contents in books and decision-making trajectories in MDP. To demonstrate this process in detail, we use the football environment as a case study. We trace how $\mathbf{M}_\pi$ outputs an imaginary action $\hat{a}$ on an imaginary state $\hat{s}$, which is actually collected at the start of a football game in GRF. We record all the intermediate outputs to trace the action generation process. The result is shown in Fig. 5. It is clearly shown that the output action "dribble" is fully supported by the logic in "has_space" branch in $K_I$, "assess_forward_decisions" function in $K$, and the paragraph of "Player in possession of the ball" in the raw content of tutorial books. However, we would like to point out that there are also some

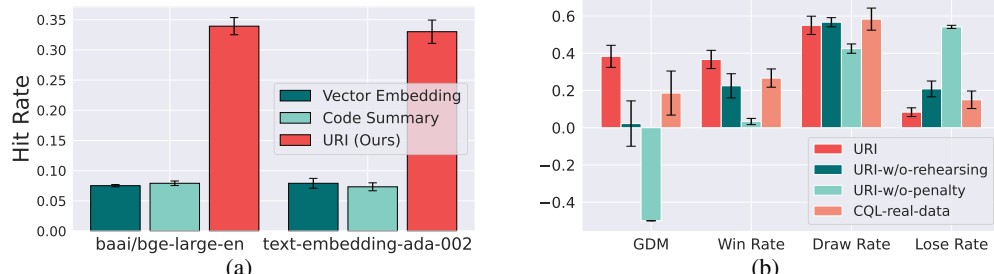

**Figure 4:** (a). Comparison of different code retrieval methods on two pre-trained language models. (b). Performance comparison of different variants of the URI framework in the GRF. This figure illustrates the average GDM, win, draw, and lose rates among the three levels of built-in AIs. The error bars in the figure indicate the standard deviation from the mean performance for each configuration in three random seeds.

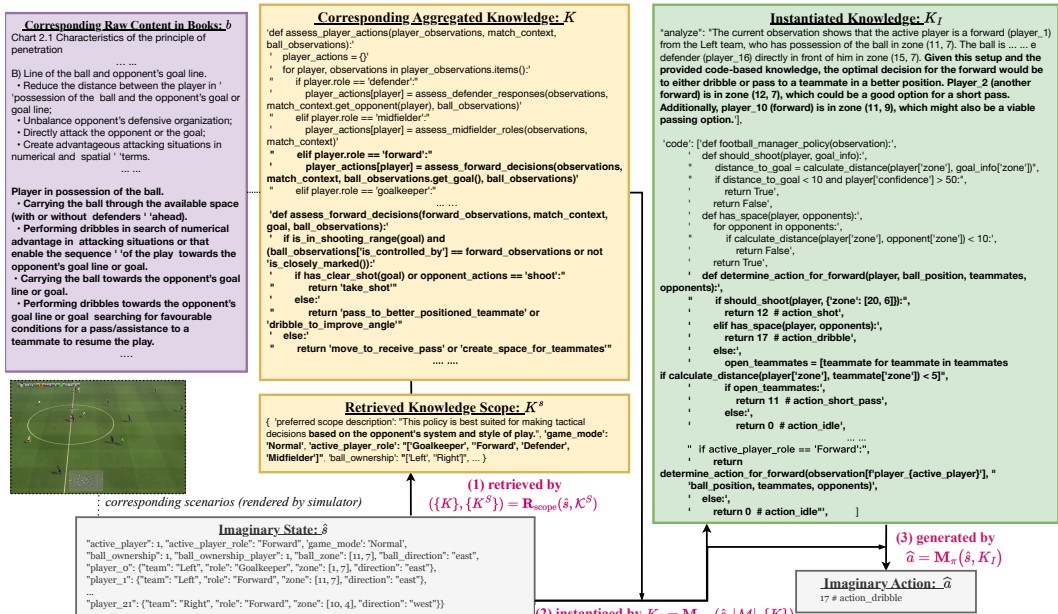

**Figure 5:** An example of the URI data generation process in the football game. The imaginary state $\hat{s}$ is collected from the real environment, which is 12 timesteps of a football game, while the rendered image is the corresponding scenario generated by the simulator. The imaginary action is "dribble", where the logic is supported by the "has_space" branch in $K_I$. Based on this, we **bold** the relevant information in the predecessor nodes and skip irrelevant information with the ellipsis "... ...".

examples that demonstrate LLMs having hallucinations in the generation, and the retrieved module might also miss the ground-truth piece of knowledge. More results are provided in Appendix H. These results indicate the necessity of introspecting in URI. The relevant ablation studies are in Sec. 6.6.

### 6.6 Importance of the components in URI

We validate the effectiveness of the rehearsing technique presented in Sec. 5.3 and the CIQL method introduced in Sec. 5.4 through ablation studies. Specifically, we constructed the following variants of the URI framework in GRF task: (1) **URI-w/o-rehearsing**, which solves the policy without using the rehearsing dataset and relies on a pre-collected dataset of 7,500 samples for offline RL policy training; (2) **URI-w/o-penalty**, where penalties $\eta_T$ and $\eta_R$ are zero, same as the standard CQL for policy learning; (3) **CQL-real-data**, where we collect real data of equivalent scale to the rehearsing-generated data using rule-based AI and apply standard CQL for offline RL policy training. The results are shown in Fig. 4(b).

Firstly, URI-w/o-rehearsing demonstrates that without generating a substantial amount of imaginary data through rehearsing, solely relying on offline RL algorithms to train a policy with our pre-collected 7,500 samples is ineffective. The results indicate that it cannot even beat the AI on Easy difficulty, though it still performs better than a random strategy. URI-w/o-penalty underscores the importance

of penalizing the uncertain aspects of the outcomes generated from the imaginary data. Neglecting this penalty leads to results worse than those of URI-w/o-rehearsing. Finally, our method slightly outperforms CQL-real-data. We attribute this improvement to the strategies inferred from prior knowledge by LLMs, which are partially superior to built-in AI behaviors, or possibly because LLMs generate more diverse data. The successful ablation results in such a challenging domain illustrate the considerable potential of PLfB.

## 6.7 Efficiency in inference

Lower cost of inference is one of the benefits for agents controlled by URI policies. We compare the inference time per action for different methods in the GRF task in Tab. 3. Our approach takes only 0.009 seconds on average to choose an action, which is significantly faster at least 300 times than using LLMs directly as agents (2.84 seconds) or with retrieval-augmented generation

**Table 3:** Comparison of inference time per action for different methods in GRF.

|  | Time Cost(s) |
| --- | --- |
| LLM-as-Agent | $2.84 \pm 0.71$ |
| LLM-RAG | $4.12 \pm 1.46$ |
| URI | $\mathbf{0.009 \pm 0.0004}$ |

(RAG) (4.12 seconds) in complex environments. This makes URI more suitable for real-time decision-making in the football simulator. The efficiency gain comes from the fact that URI distills the knowledge from the LLM into a compact policy network, which can be executed quickly without the need for expensive LLM inference at each step.

## 6.8 Imaginary Dataset Visualization

We visualize the imaginary datasets to analyze the quality of generation and uncertainty estimation. Here we present the results from the GRF, while the TTT results can be found in Appendix G.1. We choose t-SNE [59] as the visualization method and project the imaginary dataset and a real dataset collected by the rule-based policy into 2-d space for comparison. The results are in Fig. 6. The "real data" marks the data collected by the rule-based policy, while "low-unc. data" and "high-unc. data" represent segments of the imaginary dataset categorized by their uncertainty scores $R_T$ and $R_R$ falling within the lower and upper 50% percentiles, respectively. The real data and the imaginary data follow a similar data distribution, which indicates the effectiveness of the rehearsing process in URI. Besides, as highlighted by the yellow dashed circles, the uncertainty score also identifies parts of the clusters that are out of

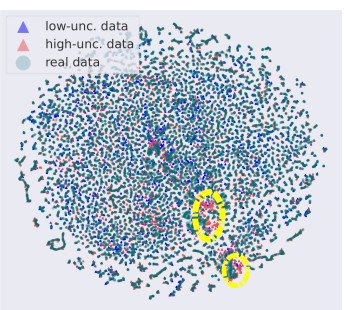

**Figure 6:** Visualization of the projected distributions for real and imaginary datasets in the football environment.

the real data distribution, which will be penalized when introspecting via CIQL.

## 7 Conclusion and Discussion

Inspired by the learning behavior of humans when they try to acquire a new skill, we propose PLfB that trains an agent from books, instead of numerous interaction data with the real environment. We also implement a practical algorithm of understanding, rehearsing, and introspecting modules to realize such a learning paradigm. The result of deploying our method in Tic-Tac-Toe and football game environments demonstrates a huge improvement in the winning rate over the baseline methods. This success proves the feasibility of utilizing knowledge stored in various written texts for decision-making agents, which was neglected by the community for a long time.

One major limitation of URI is its implicit dependence on the quality of the "tutorial books": they should cover sufficiently the dynamics, policy, and rewards of the targeted environment so that relevant knowledge and imaginary data can be extracted to train the policy. It is also important to develop metrics evaluating the textual data quality to decide whether to use URI.

Besides, we hope that the promising result in current experiments will initiate more research on PLfB. We leave more discussion of several interesting open problems in PLfBthat URI does not address in Appendix I.

## Acknowledgments and Disclosure of Funding

This work is supported by the Jiangsu Science Foundation (BK20243039). The authors thank anonymous reviewers for their helpful discussions and suggestions for improving the article.

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

# Appendix

## Table of Contents

# A  Tic-Tac-Toe (TTT) Task Setup

## A.1  Environment

Tic-Tac-Toe is a classical two-player game played on a 3×3 grid. Players take turns placing their symbols (X or O) in empty cells, with the goal of forming a line of three marks horizontally, vertically, or diagonally. The game state is represented by a 9-dimensional vector, where each dimension corresponds to a cell that can be empty (0), marked by X (1), or marked by O (2). The action space consists of integers from 0 to 8, representing the cell indices for placement.

## A.2  Tutorial Text Generation

### A.2.1  Data Collection

We first collect comprehensive game trajectories by having a minimax player (optimal policy) play against all possible opponent moves. The data collection process:

1. Start from empty board
2. For X's moves (minimax player):
   - Use optimal minimax strategy
   - First move randomized for diversity
3. For O's moves (opponent):
   - Explore all possible valid move
   - Record complete trajectory
4. Store trajectories with:
   - Complete state sequence
   - Actions taken
   - Player turns
   - Game outcomes
   - Winning/losing patterns

This systematic exploration generates a dataset covering all possible game scenarios under optimal play.

### A.2.2  Knowledge Extraction

We use GPT to analyze these trajectories and extract strategic knowledge. The prompt are used as follow:

---

**Prompt 1: Knowledge Extraction Prompt**

Given a Tic-Tac-Toe game trajectory where X is known to be the oracle (optimal player), analyze the game and provide extracted knowledge about various aspects:

1. Observe the entire game trajectory, including moves by both X and O.

2. Formulate concise statements of knowledge that capture insights about:

- Provided Trajectory Analysis: Overall assessment of game flow and key decisions
- Learned Game Mechanics: Turn-taking, move validity, board state changes
- Learned Winning Conditions: Victory achievement and prevention
- Learned Strategic Principles: General strategies demonstrated by optimal player

Your output should consist of 3-4 concise statements, each focusing on one aspect. These statements should be generalizable and applicable broadly.

---

> Example output:
>
> - Trajectory Analysis: "This game showcased a defensive strategy by X, consistently blocking O's potential winning moves while building towards a win."
> - Game Mechanics: "Tic-Tac-Toe operates on alternating turns, with each player placing their symbol in empty cells."
> - Winning Conditions: "Achieving three symbols in a line secures victory, highlighting both offensive and defensive needs."
> - Strategic Principles: "Controlling the center provides the most potential winning lines."
>
> Please provide 3-4 such knowledge statements based on the specific game trajectory presented.

### A.2.3 Generated Knowledge

The complete knowledge base is available at: `https://github.com/ziyan-wang98/URI_video_NeurIPS/blob/main/tic_tac_toe_knowledge.jsonl`

Representative examples include:

1. Strategic Principles:

   "Prioritize creating two potential winning lines simultaneously to force the opponent into a defensive position and increase chances of victory."

2. Position Control:

   "Control the center and occupy adjacent positions to create multiple threats while forcing opponent into defensive posture."

3. Opening Strategy:

   "Claim the center early when playing X to maximize board control and create multiple winning opportunities."

4. Tactical Patterns:

   "When establishing dominance, prioritize claiming center and a row/column to establish multiple winning opportunities while blocking opponent threats."

This knowledge base provides complete coverage of optimal strategies, serving as ground truth for evaluating PLfB's ability to learn from textual instructions.

## B  Google Research Football (GRF) Tasks Setup

In this section, we mainly focus on the GRF implementation[2] and the modifications we made under this environment.

### B.1  State Space

The raw state of each player contains information about the game state, the ball and all players on the pitch. The game state includes scores, game modes indicating free kicks, corner kicks or other game stages, and game time represented as steps. The ball information includes its spatial position, direction, and an indicator for ball ownership (identifying the player and team that possess it). The player information comprises position, direction, roles, yellow card records, and more. For players on the opposite team, positions and directions are mirrored. The details of the raw state are as follows:

---

[2]https://github.com/google-research/football/blob/master/gfootball/

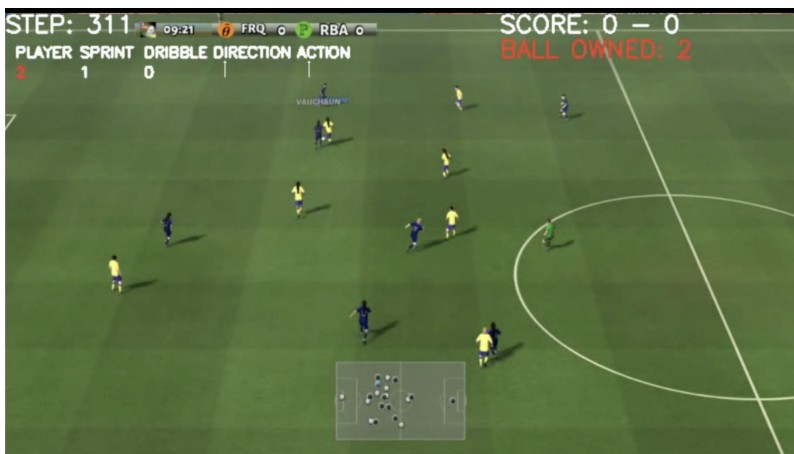

**Figure 7:** Illustration of the game in football simulator.

- **Ball Information:**
  - ball – $[x, y, z]$ position of the ball.
  - ball_direction – $[x, y, z]$ ball movement vector.
  - ball_rotation – $[x, y, z]$ rotation angles in radians.
  - ball_owned_team – {-1, 0, 1}, where -1 indicates the ball is not owned, 0 denotes the left team, and 1 the right team.
  - ball_owned_player – {0..N-1} integer denoting the index of the player owning the ball.
- **Left Team:**
  - left_team – N-elements vector with $[x, y]$ positions of players.
  - left_team_direction – N-elements vector with $[x, y]$ movement vectors of players.
  - left_team_tired_factor – N-elements vector of floats in the range {0..1}. 0 means the player is not tired at all.
  - left_team_yellow_card – N-elements vector of integers denoting the number of yellow cards a given player has (0 or 1).
  - left_team_active – N-elements vector of booleans denoting whether a given player is playing the game (False means the player got a red card).
  - left_team_roles – N-elements vector denoting roles of players, where:
    * 0 = e_PlayerRole_GK - goalkeeper,
    * 1 = e_PlayerRole_CB - centre back,
    * 2 = e_PlayerRole_LB - left back,
    ...
- **Right Team:** Same attributes as for the left team.
- **Controlled Player Information:**
  - active – {0..N-1} integer denoting the index of the controlled players.
  - designated – {0..N-1} integer denoting the index of the designated player.
  - sticky_actions – 10-elements vectors of 0s or 1s denoting whether a corresponding action is active.
- **Match State:**
  - score – Pair of integers denoting the number of goals for the left and right teams, respectively.
  - steps_left – How many steps are left till the end of the match.
  - game_mode – Current game mode.

**For the imaginary dataset generation, we created our own Imaginary state**, which includes the following information:

- **Game Information:**

- **Sticky actions:** A list of currently active sticky actions.

- **Game mode:** The current game mode.

- **Score:** The current score of the game.

- **Time:** The current game time in minutes and seconds.

- **Active player:** The index of the currently active player.

- **Active player role:** The role of the currently active player.

- **Ball ownership:** The team currently in possession of the ball (none, left team, or right team).

- **Ball ownership player:** The index of the player currently in possession of the ball.

- **Ball zone:** The zone where the ball is located.

- **Ball direction:** The direction in which the ball is moving.

- **Player Information:**

  - **Team:** The team the player belongs to (left team or right team).

  - **Role:** The role of the player.

  - **Zone:** The zone where the player is located.

  - **Direction:** The direction the player is facing.

The imaginary state is designed to provide a more compact and informative representation of the game state compared to the raw state. It extracts and processes the relevant information from the raw state, making it easier to understand and use for generating imaginary data.

The sticky actions, game mode, score, time, active player, and active player role provide a high-level overview of the current game state. The ball ownership and ball ownership player indicate which team and player are currently in control of the ball. The ball zone and direction give spatial information about the ball's location and movement.

For each player, the Imaginary state includes their team, role, zone, and direction. This information helps in understanding the positioning and orientation of the players on the pitch.

By including these key pieces of information in the Imaginary state, we aim to capture the essential aspects of the game state that are relevant for generating realistic and diverse imaginary data. The imaginary state serves as a preprocessed and structured representation of the raw state, making it more suitable for the subsequent steps in the imaginary data generation process.

Here, we show an example of imaginary state,

```
Imaginary State

{'active_player': 5,
 'active_player_role': 'Defender',
 'ball_direction': 'west',
 'ball_ownership': 2,
 'ball_ownership_player': 18,
 'ball_zone': [15, 9],
 'game_mode': 'Normal',
 'player_0': {'direction': 'east',
             'role': 'Goalkeeper',
             'team': 'Left',
             'zone': [2, 7]},
 'player_1': {'direction': 'east',
             'role': 'Forward',
             'team': 'Left',
             'zone': [12, 11]},
 'player_10': {'direction': 'east',
              'role': 'Forward',
              'team': 'Left',
              'zone': [16, 4]},
 'player_11': {'direction': 'southwest',
              'role': 'Goalkeeper',
              'team': 'Right',
              'zone': [19, 7]},
 'player_12': {'direction': 'west',
              'role': 'Forward',
              'team': 'Right',
              'zone': [14, 4]},
 'player_13': {'direction': 'north',
              'role': 'Forward',
              'team': 'Right',
              'zone': [14, 8]},
 ...
 'score': [0, 0],
 'step': 295,
 'sticky_actions': ['BottomRight', 'Sprint'],
 'time': '8 minutes 51 seconds'}
```

To enhance the understanding of player and ball positions beyond mere coordinates, our methodology involves dividing the football field into a grid of 20 x 12 rectangular **zones**. This granular zoning approach serves a dual purpose: firstly, it abstracts the complex spatial dynamics of the game into a more manageable form, allowing our LLM to process and generate data with increased effectiveness. Secondly, it provides a framework for more accurately simulating the spatial strategies and movements employed in real-world football scenarios. Each zone, defined by specific dimensions, acts as a unique spatial identifier, offering precise reference points for player positioning and ball location. This system not only simplifies the representation of the field but also enriches the strategic depth of the game model, as actions and decisions can be tailored to the distinct characteristics of each zone. By adopting this zoning strategy, we aim to bridge the gap between the high-level strategic understanding required for football and the detailed, positional awareness needed to implement those strategies effectively within the simulated environment.

The simulator information $|\mathcal{M}|$ includes the description of the state space $\mathcal{S}$. What we use in this paper is as follows.

- First, it provides information such as the time and score of the match.

- Second, which side has control of the ball, and the active player in the left team, you need to propose corresponding policies to him.

- Next, we present the position and role information of each player: In this text description, the football grass field is divided into 240 zones.
- We use the zone (x, y) to express the position of the player. "x" is the distance from the left team's penalty area to the right team's penalty area, ranging from 1 to 20, and y is the distance from the lower corner to the upper corner flag, ranging from 1 to 12. This means that the center circle position of the field is zone (10, 6), where the game start.
- The lower left corner position of the left team is (1, 1), and the upper right corner position of the right team is (20, 12).
- The venues never interchange or change. The direction of the position information is the direction the player is currently facing and the direction of future actions.

## B.2 Action Space

The default action set comprises 19 actions, including directional movements, three various ball passing, ball shooting, sliding, sprinting and others. Throughout our experiments, we utilized the default action set. In details,

- **Idle actions**
  `action_idle = 0` a no-op action, sticky actions are not affected (player maintains his directional movement etc.).
- **Movement actions**
  `action_left = 1:` run to the left, sticky action.
  `action_top_left = 2:` run to the top-left, sticky action.
  `action_top = 3:` run to the top, sticky action.
  `action_top_right = 4:` run to the top-right, sticky action.
  `action_right = 5:` run to the right, sticky action.
  `action_bottom_right = 6:` run to the bottom-right, sticky action.
  `action_bottom = 7:` run to the bottom, sticky action.
  `action_bottom_left = 8:` run to the bottom-left, sticky action.
- **Passing / Shooting**
  `action_long_pass = 9:` perform a long pass to the player on your team. Player to pass the ball to is auto-determined based on the movement direction.
  `action_high_pass = 10:` perform a high pass, similar to `action_long_pass`.
  `action_short_pass = 11:` perform a short pass, similar to `action_long_pass`.
  `action_shot = 12` perform a shot, always in the direction of the opponent's goal.
- **Other actions**
  `action_sprint = 13:` start sprinting, sticky action. Player moves faster, but has worse ball handling.
  `action_release_direction = 14:` reset current movement direction.
  `action_release_sprint = 15:` stop sprinting.
  `action_sliding = 16:` perform a slide (effective when not having a ball).
  `action_dribble = 17` : start dribbling (effective when having a ball), sticky action. Player moves slower, but it is harder to take over the ball from him.
  `action_release_dribble = 18:` stop dribbling.

## B.3 Transition

The game dynamics in GRF closely resemble realistic football games. Players can move or sprint with or without the ball. They can also execute various passes and shots. Besides different actions, complex interactions such as collisions and trips between players are simulated as well. Additionally, the game engine introduces stochasticity in the dynamics, which can influence the passes and shots randomly.

In our experiment, to address the challenge of executing fine-grained maneuvers within a game environment, we employ a strategy where actions generated by rule-based policies [58] are utilized

to replace those produced by the policies, including URI and all of the baselines. This substitution is particularly enacted when considering the spatial segmentation of the play area into zones. Using zones as a pivotal input, the approach facilitates more nuanced control over in-zone activities. The rationale behind this decision is rooted in the state that precise actions within these designated zones can significantly impact the outcome of play, necessitating a method that allows for detailed operational control. This method ensures that the agents can perform more sophisticated strategies, particularly in scenarios that demand high levels of precision and situational awareness within the confined spaces of each zone.

# C    Additional Related Work

Offline RL addresses the problem of learning policies from a pre-collected dataset. Most methods can be classified into two categories: model-free and model-based methods. Model-free [40–43, 5, 44, 45] methods learn a conservative policy directly from the dataset. Model-based offline algorithms [6, 46–49] first estimate a generalizable dynamics model [60, 52] from the dataset and perform policy learning or planning based on this learned model. In our work, to achieve introspection from the data generated by the LLMs, we build our policy distillation algorithm based on several existing techniques in offline RL, including the uncertainty penalty in MOPO [6] , which constructs a pessimistic model that discourages the policy from visiting states where the model is inaccurate; and conservative Q-learning loss [5] to obtain a robust value function that does not overestimate unseen state-action pairs too much.

# D    Baseline Implementations for TTT

In this section, we detail the implementation of baselines for both Tic-Tac-Toe.

## D.1    LLM-as-agent

The LLM-as-agent baseline uses GPT-3.5 to directly generate actions for both environments. we use the following prompts:

---

**Prompt 2: LLM-as-agent Global Prompt for Tic-Tac-Toe**

\# Tic-Tac-Toe

\#\# Introduction

Tic-Tac-Toe is a classic two-player game, unfolds on a 3-by-3 grid where the objective is to align three of one's symbols, Xs for the first player and Os for the second, either horizontally, vertically, or diagonally. Strategic placement is crucial: besides aiming for three in a row, players must also block their opponents' potential alignments to avoid defeat. Players can place their next move in an empty cell on the 3-by-3 grid.

\#\# Rules

1. The game is played on a 3x3 grid.
2. Players take turns placing their symbol (X or O) in empty cells.
3.  The first player to get 3 of their symbols in a row (horizontally, vertically, or diagonally) wins.
4. If all cells are filled and no player has won, the game is a draw.

\#\# Game State Representation

The game board is represented as a string of 9 characters, where:
- 'X' represents the first player's moves
- 'O' represents the second player's moves
- ' ' (space) represents an empty cell

---

## Your Task:

- Respond with a JSON object containing your move and reasoning.

- The move should be a single integer from 0 to 8, representing the cell index.

- Provide a brief explanation for your choice in the reasoning field.

---

**Prompt 3: LLM-as-agent Query Prompt for Tic-Tac-Toe**

## Choose the next move

You are playing as '{player}'.

The current state of the board is as follows:

state.board[0] | state.board[1] | state.board[2]
————————————————————————-
state.board[3] | state.board[4] | state.board[5]
————————————————————————
state.board[6] | state.board[7] | state.board[8]

The next move should be placed in an empty cell on the 3-by-3 grid.

The available positions for the next move are:

{state.available_moves()}

Positions are represented by the numbers 0-8, corresponding to the cell's index:

0 | 1 | 2
————
3 | 4 | 5
————
6 | 7 | 8

Analyze the game state and choose your next move. Respond with a JSON object in the following format:
{ "move": <integer 0-8>, "reasoning": "<brief explanation for your choice>" }

---

### D.2 LLM-RAG

The LLM-RAG baseline enhances LLM-as-agent by retrieving relevant knowledge from tutorial books.

---

**Prompt 4: LLM-RAG Global Prompt for Tic-Tac-Toe**

**[Same as LLM-as-agent Global Prompt with additional instruction:]**

The policy knowledge from tutorials will be provided to guide your decision.

> **Prompt 5: LLM-RAG Query Prompt for Tic-Tac-Toe**
>
> The relevant policy knowledge for the current state is:
>
> {policy_knowledge}
>
> ## Choose the next move based on the policy knowledge
>
> **[Rest same as LLM-as-agent Query Prompt]**
>
> You should follow the provided policy knowledge to make your decision.
>
> Base your reasoning on how your chosen move aligns with the strategic principles described in the policy knowledge.

For the RAG part, we use the following knowledge retrieval approach:

- Use GPT embeddings to encode both state descriptions and tutorial knowledge
- Retrieve top-3 most relevant pieces of strategic knowledge using cosine similarity
- Combine retrieved knowledge with state information for move generation

## D.3 Additional Baselines

- **Minimax (Optimal)**:
  - Uses standard minimax algorithm with perfect play
  - Evaluates states with +1 for win, -1 for loss, 0 for draw
  - Uses random first move selection for diversity
- **Minimax-noise**:
  - Based on optimal minimax algorithm
  - Makes random moves with 30% probability
  - Uses optimal moves with 70% probability
  - Serves as a strong but imperfect opponent
- **Random**:
  - Selects moves uniformly from available positions
  - Serves as a baseline for minimal strategic play

# E  Baseline Implementations for GRF

In this section, we will focus on implementing of two relevant baselines.

## E.1  LLM-as-agent

As mentioned in the paper, the LLM-as-agent method uses LLM (GPT3.5) directly to generate the action. We provide the prompt as follows,

> **Prompt 6: LLM-as-agent Query Prompt 1/2**
>
> The texted state for the current state:
> {text_state}
> ——————————
> {Simulator Information $|\mathcal{M}|$. Refer to Appendix F.4.}
> ——————————
>
> For example, if the active player is Player 2 and you want him to be close to the ball and control it in the texted state,

- Forward Player 2 is at Zone(9,9).
- The ball is at Zone(11,8).

Given these coordinates, the ball is diagonally one zone to the right (east) and one zone down (south) from the player's current position. The most direct route to the ball would indeed be diagonally towards the bottom right.

Therefore, the most appropriate action for Forward Player 2 in this situation would be:

6 = action_bottom_right: This sticky action will allow the player to move diagonally in the bottom-right direction (southeast), which is the direct path to where the ball is currently located in Zone(11,8).
By choosing action_bottom_right, Forward Player 2 can close the distance to the ball more effectively, aligning their movement directly with the ball's current location. Once the player reaches the ball, the subsequent action can be decided based on the situation at that moment (e.g., dribbling, passing, or shooting).

______________

Question: What next action do you want this active player to take?
Answer:

The global prompt shows as follows,

## Prompt 7: LLM-as-agent Global Prompt

I want you to act like a football manager and also an expert in Python coding and Reinforcement Learning, which is about learning a football manager policy in a football simulator.

I will give you the pseudocode snippets to define the specific policy for the active player in the football game.

These codes represent absolute correctness and do not add other common logic. Your task is to select the action that best fits and executes the code based on the logic of the given code.

**Requirements:**

**About the action choosing:**
- Please choose the action that best fits the code logic.

**About the format:**
- you should answer in pure JSON format with the key: 'action': an int number from 0 to 18,
- 'thought': why did you choose this action? without any other information or code.
- For example, you should not add the "'JSON'" tag in the answer.

Response example (you should respond in the following order):

{{

"action": 0,

"thought": "Based on your thought, tell me the optimal action you would like to select in the action set."

```
}}
```

## E.2  LLM-RAG

The LLM-RAG enhances LLM-as-agent by retrieving relevant knowledge from the database extracted from the tutorial books, similar to the retrieval step in our rehearsing stage, but directly outputs the action without policy learning. The dataset which the LLM-RAG using is the text-booked dataset. Here we show the prompts of LLM-RAG.

---

**Prompt 8: LLM-RAG Query Prompt 1/2**

The relative policy from the book you want this active player to implement is as follows:

    {Policy_Str}
___________________

The texted state for the current state:

    {text_state}
___________________
{Simulator Information $|\mathcal{M}|$. Refer to Appendix F.4.}
___________________

For example, if the active player is Player 2.

The most relative policy from books:

- When you become a back defender, you aim to get close to the ball and stop the attacker's progression towards the goal.

The texted state:

- Forward Player 2 is at Zone(9,9).
- The ball is at Zone(11,8).

Given these coordinates, the ball is diagonally one zone to the right (east) and one zone down (south) from the player's current position. The most direct route to the ball would indeed be diagonally towards the bottom right.

Therefore, the most appropriate action for Forward Player 2 in this situation would be:

6 = action_bottom_right: This sticky action will allow the player to move diagonally in the bottom-right direction (southeast), which is the direct path to where the ball is currently located in Zone(11,8).
By choosing action_bottom_right, Forward Player 2 can close the distance to the ball more effectively, aligning their movement directly with the ball's current location. Once the player reaches the ball, the subsequent action can be decided based on the situation at that moment (e.g., dribbling, passing, or shooting).

___________

Question: What next action do you want this active player to take?
Answer:

---

The global prompt shows as follows,

> **Prompt 9: LLM-RAG Global Prompt**
>
> I want you to act like a football manager and also an expert in Python coding and Reinforcement Learning, which is about learning a football manager policy in a football simulator.
>
> I will give you the pseudocode snippets to define the specific policy for the active player in the football game.
>
> These codes represent absolute correctness and do not add other common logic. Your task is to select the action that best fits and executes the code based on the logic of the given code.
>
> **Requirements:**
>
> **About the action choosing:**
>
> - Please choose the action that best fits the code logic.
>
> **About the format:**
>
> - you should answer in pure JSON format with the key: 'action': an int number from 0 to 18,
> - 'thought': why did you choose this action? without any other information or code.
> - For example, you should not add the "'JSON'" tag in the answer.
>
> Response example (you should respond in the following order):
>
> {{
>
> "action": 0,
>
> "thought": "Based on your thought, tell me the optimal action you would like to select in the action set."
>
> }}

For the RAG part, we use the llama index [61][3] for embedding the textbook paragraphs and the state and find the most relevant one as input to the query prompt.

# F   URI Implementation

In this section, we will focus on implementing URI.

## F.1   Book Content Understanding

Book content understanding is achieved via a code-based knowledge extractor and a code-based knowledge aggregator. The implementation is designed to be task-agnostic, with task-specific parameters configured through prompts.

### F.1.1   Knowledge Extractor

The knowledge extractor processes book paragraphs to identify and extract relevant knowledge. Algorithm 1 shows the extraction process:

---

[3]https://github.com/run-llama/llama_index

**Algorithm 1** Code Extractor

**Require:** Book $\mathcal{B}$ consisting of segments $b_1, b_2, \ldots, b_{N_b}$
1: Initialize language model $\mathbf{M}_{\text{ext}}$
2: Define KnowledgeContext $\leftarrow$ {Rewards, Policies, Dynamics}
3: $\mathcal{K} \leftarrow \emptyset$
4: $\mathcal{I} \leftarrow \emptyset$
5: **for** $b_i \in B$ **do**                            ▷ Iterate over each paragraph in the books
6:      $\mathcal{I}.\text{add}(b_i)$
7:      $Output \leftarrow \mathbf{M}_{\text{ext}}(\mathcal{I}, \text{KnowledgeContext})$
8:      **if** $'\text{WithKnowledge}'$ in $Output$ **then**
9:          $\mathcal{K}.\text{add}(Output.code)$
10:      **else if** $'\text{WithoutKnowledge}'$ in $Output$ **then**
11:          $\mathcal{I} \leftarrow \emptyset$
12:      **end if**
13: **end for**
14: **return** $\mathcal{K}$

Task-specific configuration can be found at Appendix. F.3 and F.4

### F.1.2 Knowledge Extraction Prompt

---

**Prompt 10: Book Information Extraction Prompt**

I want you to act **{task_info}**.

You need to analyze the given paragraph step-by-step from a related context to derive the specific theorem, principle, rule, and law of the related elements or concepts:

**{element_type_prompt_dict[knowledge_type]}.**

**{filter_stringency_prompt}.**

Requirements:

About the answer:

- If you think the paragraph contains the above elements, please answer 1. The answer is 1 only when you can write the specific theorem, principle, rule, and law of the related elements into pseudocode snippet.
- If you think the paragraph does not contain the above elements, please answer 0.
- If you think the given paragraph is not clear enough to answer, please answer 2. Then I will give you the following paragraph to help you answer.
- If you think the paragraph contains the above elements but the content is not clear enough to derive the specific theorem, principle, rule, and law of the related elements, please answer 2. Then I will give you the following paragraph to help you answer.

About the analysis:

- If the answer is 1, you should give the specific theorem, principle, rule, and law of the related elements.
- You should write the specific theorem, principle, rule, and law of the related elements via pseudo code.
- Please provide the PYTHON-style pseudo code as detailed as you can to cover the most information of the original content.

---

About the pseudo code snippet:

- **{element_type_code_example_dict[knowledge_type]}**

About the format:

- You should answer in pure JSON format, without any other information or code.

Response example:

```
{
"answer": 1,
"pseudocode of/about XX": ["PYTHON pseudocode snippet1", ...],
"pseudocode of/about XX": ["PYTHON pseudocode snippet2", ...]
}
```

### F.1.3 Knowledge Aggregation

The aggregation process combines similar pieces of knowledge into more concise representations. Algorithm 2 shows the process:

---

**Algorithm 2** Code-based Knowledge Aggregation (one round)

---

**Require:** Knowledge $\mathcal{K}$, text inclusion flag $T$, aggregation number $N$, similarity threshold $\tau$
1: Initialize language model $\mathbf{M}_{\text{agg}}$ and embedding model $\mathbf{E}$
2: $\mathcal{K}' \leftarrow \emptyset$
3: **for** $K$ in $\mathcal{K}$ **do**                                     ▷ Iterate over each knowledge piece
4:     Define KnowledgeContext $\leftarrow$ {Rewards, Policies, Dynamics}
5:     $\mathcal{K}^{\text{sim}} \leftarrow \{K' \in \mathcal{K} | \text{calculate-similarities}(\mathbf{E}(K), \mathbf{E}(K')) > \tau\}$
6:     **if** $|\mathcal{K}^{\text{sim}}| \geq N_{\text{agg}}$ **then**
7:         $K^{\text{agg}} \leftarrow \mathbf{M}_{\text{agg}}(\mathcal{K}^{\text{sim}}, \text{KnowledgeContext})$
8:         $\mathcal{K}'.\text{add}(K^{\text{agg}})$
9:     **else**
10:         $\mathcal{K}'.\text{add}(K)$
11:     **end if**
12: **end for**
13: **return** $\mathcal{K}'$

---

Key parameters for aggregation:

- $N_{\text{agg}}$: Number of similar pieces to aggregate (default: 4)
- $\tau$: Similarity threshold (default: 0.95)
- embedding_model: Model for computing text similarities

### F.1.4 Knowledge Aggregation Prompt

---

**Prompt 11: Code Aggregation Prompt**

I want you to act like **{task_info}**.

I will give you several pseudocode snippets to define the specific theorem, principle, rule, and law of the related elements or concepts:

**{element_type_prompt_dict[knowledge_type]}.**
**{filter_stringency_prompt}**

---

These codes might share some common logic. Your task is to aggregate the common logic of the given codes into a single code snippet, while still covering all of the given codes.

Requirements:

About the aggregated code:

- Please provide the PYTHON-style pseudo code as detailed as you can
- Using the least number of pseudocode items
- Covering all of the code. However, please feel free to add more pseudocode items if needed
- Since I will delete the original code after getting your aggregated code, you cannot call the pseudocodes that I provided in the prompt

About the pseudo code snippet:

- **{element_type_code_example_dict[knowledge_type]}**

About the format:

- You should answer in pure JSON format

Response example:

```
{
"pseudocode of/about XX": ["PYTHON pseudocode snippet1", ...],
"pseudocode of/about XX": ["PYTHON pseudocode snippet2", ...]
}
```

---

**Prompt 12: Text-based Aggregation Prompt**

I want you to act like **{task_info}**.

I will give you several paragraphs and the corresponding summary written by code from several related books. You need to analyze the given paragraph step-by-step from a related context to aggregate the specific theorem, principle, rule, and law of the related elements or concepts:

**{element_type_prompt_dict[knowledge_type]}.**

**{filter_stringency_prompt}**

Requirements:

About the analysis:

- You should write the specific theorem, principle, rule, and law via pseudo code
- Please provide the PYTHON-style pseudo code as detailed as you can

You should aggregate the given information as much as you can:

1. Using the least number of pseudocode items
2. Covering all of the code and most of the original texts
3. Since I will delete the original code after aggregation, you cannot call the provided pseudocodes

About the pseudo code snippet:
- **{element_type_code_example_dict[knowledge_type]}**

About the format:
- You should answer in pure JSON format

Response example:

```
{
"aggregated_pseudocode of/about XX": ["PYTHON pseudocode snippet1", ...],
"aggregated_pseudocode of/about XX": ["PYTHON pseudocode snippet2", ...]
}
```

## F.2 Knowledge-based Rehearsing of Decision-Making

Knowledge-based rehearsing of decision making is achieved via a state-space knowledge scope retrieval module and retrieved knowledge instantiation.

For state-space knowledge scope retrieval, we use GPT-3.5 to identify the knowledge scope represented by state space, then adopt a standard RAG module to retrieve the correct pieces of knowledge.

---

**Prompt 13: Football Knowledge Code Generation Prompt**

I want you to act like a football manager and also an expert in python coding and Reinforcement Learning that wants to learn a football manager policy in a football simulator.

I will provide an observation space of the football simulator and some policy functions written by others. Your task is to summarize an observation scope, which is a subspace of the observation space and most suitable to use this policy function to win the game.

The observation space: **{observation_space_desc}**

The action space: **{action_space_desc}**

We have N codes for you to analyze: **{code_list}**
About the definition of the observation scope of each code:
For each code, you should return a description of the best scope of the observation variables to be used as the policy function to win the game, including:
- A text summary of the preferred/better scope
- score is preferred/better to be in [?]
- active_player_role is preferred/better to be in [?]
- ball_ownership is preferred/better to be in [?]
- ball_ownership_player is preferred/better to be in [?]
- ball_zone is preferred/better to be in [?]
- ball_direction is preferred/better to be in [?]
- {the preferred zone of all players from 0 to 21}

NOTE:
1. You should define your observation scope as detailed and tight as possible
2. You should define the observation scope for each code by comparing the differences between the codes

---

About the format:

- You should answer in pure JSON format, without any other information or code

Response example:
```
{
    'code_idx 1': {
        'preferred scope description': 'a text summary of the suitable
        situations,
        'score': 'preferred scope of the score',
        'active_player_role': 'preferred scope of the active_player_role',
        ...
    },
    ...
}
```

---

## Prompt 14: Tic-Tac-Toe Knowledge Code Generation Prompt

I want you to act like a tic-tac-toe pro player and also an expert in python coding and Reinforcement Learning that wants to learn a tic-tac-toe policy in a tic-tac-toe simulator.

I will provide an observation space of the tic-tac-toe simulator and some policy functions written by others. Your task is to summarize an observation scope, which is a subspace of the observation space and most suitable to use this policy function to win the game.

The observation space: **{observation_space_desc}**

The action space: **{action_space_desc}**

We have N codes for you to analyze: **{code_list}**

About the definition of the observation scope of each code:

For each code, you should return a description of the best scope of the observation variables to be used as the policy function to win the game, including:

- A text summary of the preferred/better scope
- Preferred board states: [(?), (?), (?), (?), (?), (?), (?), (?), (?)], where you should replace "?" with all possible values or "any" if all values are suitable
- Preferred/better player: [?], where you should replace "?" with possible values

NOTE:
1. You should define your observation scope as detailed and tight as possible
2. You should define the observation scope for each code by comparing the differences between the codes

About the format:

- You should answer in pure JSON format, without any other information or code

Response example:
```
{
    'code_idx 1': {
```

```
            'preferred scope description': 'a text summary of the suitable
            situations,
            'current_player': 'preferred scopes',
            'preferred board states': 'preferred scopes'
    },
    ...
}
```

Then a standard RAG technique is applied to identify the most relevant knowledge scope $K_j^s$ and its relevant knowledge $K_j$. Formally, $(\{K_j\}, \{K_j^S\}) = \mathbf{R}_{\mathrm{scope}}(\hat{s}, \mathcal{K}^S)$, where $\mathcal{K}^S$ is the scope of the knowledge database.

### F.2.1 Code Instantiation

After that, we employ an LLM to instantiate the code $K^I = \mathbf{M}_{\mathrm{Inst}}(\hat{s}, |\mathcal{M}|, \{K_j\})$ based on the current state $\hat{s}$, the simulator information description $|\mathcal{M}|$ and the knowledge $\{K_j\}$ retrieved by $\mathbf{R}_{\mathrm{scope}}$. The prompt is listed in Prompt 15. Since this step requires a strong understanding of the code, we use GPT-4 instead of GPT-3.5 as the LLM implementation.

The code instantiation process converts general knowledge into task-specific code through a dedicated module. Here's the detailed implementation:

---

**Prompt 15: Code Instantiation Prompt**

I want you to act **{task_info}**.

I will give you an observation which you are facing in the simulator, your task is to instantiate a code that serves as a **{knowledge_type}** function, which is used for **{knowledge_function}**.

For example, **{knowledge_function_example}**

Formally, the format of knowledge_type function is **{knowledge_format}**.

To help you complete the task, I will provide you:

1. Pieces of relevant knowledge from the tutorial, written in Python-style pseudocode
2. The observation space and action space of the target simulator
3. The current observation you are facing

Python-style relevant knowledge from tutorial: **{code_string}**

The observation space: **{observation_space_desc}** The action space: **{action_space_desc}** Current observation: **{obs}**

About your code-instantiation task:

- Please provide the PYTHON-style pseudo code as detailed as you can
- Your task is to rewrite a code that describes a **{knowledge_type}** suitable to current observation
- You should make the optimal decision based on analyzing current observation
- After analysis, rewrite the pseudocode to make it most suitable for downstream tasks
- Keep a main function named "**{knowledge_type}**" and add inner functions if necessary

---

Important Notes:
- You cannot call the pseudocodes provided in the prompt
- Variable assignment can be simplified, but implement logic in detail
- Do not use placeholder implementations

Additional Requirements:
- For one-step use: Keep code short, direct, and focused on main logic
- For multi-step use: Code should generalize to next minute of game, considering:
  - Changes in active player
  - Ball position changes
  - Opponent position changes

Response format:
```
{
    "analyze": "Analysis of current observation,
    focusing on code instantiation approach",
    "code": "Rewritten code based on current observation analysis"
}
```

### F.2.2 Rollout Process

Finally, three LLMs are involved to generate the imaginary dataset $\mathcal{D}_{\mathrm{img}}$, including $\hat{a}_t = \mathbf{M}_\pi(\hat{s}_t, \mathbf{M}_{\mathrm{Inst}}(s_t, |\mathcal{M}|, \mathbf{R}_{\mathrm{scope}}(\hat{s}, \mathcal{K}_\pi^S)))$, $\hat{r}_t = \mathbf{M}_R(\hat{s}_t, \hat{a}_t, \mathbf{M}_{\mathrm{Inst}}(s_t, |\mathcal{M}|, \mathbf{R}_{\mathrm{scope}}(\hat{s}, \mathcal{K}_R^S)))$ and $\hat{s}_{t+1} = \mathbf{M}_T(\hat{s}_t, \hat{a}_t, \mathbf{M}_{\mathrm{Inst}}(s_t, |\mathcal{M}|, \mathbf{R}_{\mathrm{scope}}(\hat{s}, \mathcal{K}_T^S)))$, where $\mathcal{K}_\pi^S$, $\mathcal{K}_R^S$, and $\mathcal{K}_T^S$ are the scope of the knowledge database $\mathcal{K}_\pi$, $\mathcal{K}_R$, and $\mathcal{K}_T$ respectively.

---

**Prompt 16: Policy Inference Prompt**

I want you to act **{task_info}**.

I will give you an observation which you are facing in a simulator, your task is to response a correct results serving as a **{knowledge_type}** function, which is used for **{knowledge_function}.**

For example, **{knowledge_function_example}.**

Formally, the format of **{knowledge_type}** function is **{knowledge_format}**.

To help you complete the task, I will provide you:
1. Pieces of relevant knowledge from the tutorial, written in Python-style pseudocode
2. The observation space and action space of the simulator
3. The current observation you are facing

Python-style relevant knowledge from tutorial books:

**{code_string}**

The observation space:

**{observation_space_desc}**

---

The action space:

**{action_space_desc}**

Current observation: **{obs}**

Requirements:
- Choose the action that best fits the code logic
- Answer in pure JSON format with keys:
  - 'action': integer from 0 to 18
  - 'thought': reasoning for the choice

Hints for inference: **{hint}**

**{inference_format_example}**

I want you to act like **{task_info}**.

I will give you an observation which you are facing in the target simulator, your task is to response a correct results serving as a dynamics function code from the tutorial.

Formally, the format of dynamics function is dynamics_format. The function describes the mechanism for updating the position of the ball and players.

To help you complete the task, I will provide you:
1. The dynamics function from tutorial books: **{dynamics_code_string}**
2. Current observation: **{obs}**
3. Chosen action: **{action}**

Hints for inference: **{hint}**

Response format:

**{inference_format_example}**

I want you to act **{task_info}.**

I will give you an observation which you are facing in a simulator, your task is to response a correct results serving as a **{reward}** function code from the tutorial.

Formally, the format of **{reward}** function is **{reward_format}**. The function describes the mechanism for calculating rewards.

Inputs provided:
1. The reward function: **{reward_code_string}**
2. Current observation: **{obs}**
3. Action taken: **{action}**
4. Next observation: **{next_obs}**

> Hints for inference: **{hint}**
>
> Response format:
>
> **{inference_format_example}**

The rollout process follows Algorithm 3, which integrates the three inference components:

---

**Algorithm 3** Knowledge-based Rollout

---

1: **function** ROLLOUTONESTEP(recent_obs, policy_code, reward_code, transition_code)
2:     text_obs ← ConvertObsToText(recent_obs)
3:     action, action_thought ← PolicyInference(text_obs, policy_code)
4:     next_obs, trans_thought ← DynamicsInference(text_obs, action, transition_code)
5:     next_text_obs ← PostProcessObs(next_obs, recent_obs, action)
6:     rewards, reward_thought ← RewardInference(text_obs, action, next_text_obs)
7:     **return** next_obs, action, rewards, thoughts
8: **end function**

---

### F.3  Tic-Tac-Toe Prompt Configurations

The following configurations detail the task-specific parameters used in the previously described prompts for the Tic-Tac-Toe environment. These include the task description, knowledge type definitions, observation space descriptions, and other parameters that are substituted into the corresponding placeholders in the book content understanding prompts (Appendix F.1), knowledge rehearsing prompts (Appendix F.2).

**{task_info}**

> "An expert Tic-Tac-Toe player and also an expert in python coding and Reinforcement Learning."

**{knowledge_type}**

- **Policy Function:**

  > "The policy in Tic Tac Toe involves strategic placement of X's or O's on the game board to block the opponent's moves and create opportunities to complete a line. For example, "Always place your mark in the center if it is your first move, as it maximizes potential winning combinations."

- **Environment Dynamics Function:**

  > "This function describes the rules and changes in the game environment following player actions. For example, "If a player completes a row, column, or diagonal with the same symbol, they win the game. Otherwise, the game continues or ends in a draw if all spaces are filled without a winner."

- **Rewards Function:**

  > "This function outlines the incentives or penalties based on the game outcomes in Tic Tac Toe. For example, "Completing three consecutive symbols horizontally, vertically, or diagonally results in a win and positive reward, while failing to block an opponent's winning move results in a loss and a negative reward."

**{knowledge_function_example}**

- **Policy Examples:**

  > "A common opening strategy is to place the first mark in the center of an empty board, which provides the most opportunities for setting up wins in subsequent

moves. Another strategy involves placing the second mark in a corner if the center is already occupied by the opponent."

- **Dynamics Examples:**

    "The board state changes with each player's move, leading to new potential lines for winning. A player wins by placing three of their marks in a horizontal, vertical, or diagonal row. If all nine squares are filled and no player has three in a row, the game is a draw."

- **Reward Examples:**

    "You can classify the outcomes as follows: 2: Winning move; 1: Blocking an opponent's winning move; 0: Neutral move; -1: Missing an opportunity to block an opponent's winning move; -2: Making a move that leads directly to a loss."

## {observation_space_desc}

"The observation space consists of a 3×3 grid representing the game board. Each cell can be empty or contain an 'X' or 'O' symbol. We use a 1×9 array with number 0, 1, and 2 to represent empty, 'X', and 'O' respectively. The board state is updated after each player's move, reflecting the current game configuration. We will use another line to tell you whose turn it is, 1 for X and 2 for O."

## {action_space_desc}

"The action space is discrete from 0 to 8, including 9 possible moves corresponding to the 9 cells on the game board. Players can place their mark ('X (1)' or 'O (2)') in any empty cell to make their move."

## {inference_format_example}

- **Policy Inference:**

    "For example, if it is X's turn and you observe the following board state: [1, 2, 0, 0, 0, 0, 0, 0, 0]
    The strategic thought might be "Placing X in the center (cell 4) could block O's potential diagonal completion and creates multiple opportunities for X to win on subsequent turns. Therefore, the most appropriate action for X in this situation would be to place in the center."
    Expected output format:

    ```
    {
        "thought": "Given the current board state
         [0,1,2,0,0,0,0,0,0] and that it's X's turn...",
        "action": 4
    }"
    ```

- **Dynamics Inference:**

    "Important keys in the observation:
    – `board`: Current board configuration as a list of nine integers (0 for empty, 1 for X, 2 for O). Example: [0,1,2,0,0,0,0,0,0]
    – `player_turn`: Indicates current player (1 for X, 2 for O)
    Expected output format:

    ```
    {
        "thought": "Given the board state [0,1,2,0,0,0,0,0,0]
        and the action to place X in the center, the next
        state of the board should be [0,1,2,0,1,0,0,0,0]
        reflecting X's placement.",
        "board": [0,1,2,0,1,0,0,0,0],
        "player_turn": 2  # O's turn next
    }"
    ```

- **Reward Inference:**

    "Based on current state, action, and resulting state, compute rewards as:
    - +2: Winning move
    - +1: Blocking opponent's winning move
    - 0: Neutral move
    - -1: Missing opportunity to block
    - -2: Move leading to direct loss

    Expected output format:

    ```
    {
        "thought": "Given the board state [0,1,2,0,0,0,0,0,0],
        the action to place X in the center, and the observed
        next state [0,1,2,0,1,0,0,0,2]. With X's placement
        resulting in no immediate win or loss and the game
        still in play, the reward for this action should be 0.",
        "dense_reward": 0
    }"
    ```

## F.4 Football Prompt Configuration

Here we detail the configuration for the Football:

**{task_info}**

"A football manager and also an expert in python coding and Reinforcement Learning"

**{knowledge_type}**

- **Policy Function:**

    The football manager policy is to give the tactics and strategies for all players in the team, such as how players should be used in the frontcourt during a match, or when forwards should take shots. For example: "When watching defenders you have to assess how they respond to their opponents as well as the ball."

- **Environment Dynamics Function:**

    Dynamics is to give the dynamics function or related rules of the football game under the football manager policy's action, such as after shotting, the ball will be in the goal or not. For example: "When the direction of shotting is vertical to the goal, the ball will be easy to the goal."

- **Rewards Function:**

    Reward is to give the reward or punishment of the football manager policy. For example: "When the forwards are restricted, the midfielder can support and take away the defenders, which is a very encouraging behavior."

**{knowledge_format}**

- **Policy Function:**

    Policy function: observation -> action

- **Environment Dynamics Function:**

    Dynamics function: observation, action -> next_observation

- **Rewards Function:**

    Reward function: observation, action, next_observation -> reward

**{knowledge_function_example}**

- **Policy Function:**

  "4-4-2 is a good formation for a team with a strong midfield, because it allows the team to control the ball and keep possession. To play this formation, the team should have two central midfielders, two wingers, and two strikers. The central midfielders should be able to pass the ball well and control the game. The wingers should be able to run up and down the wing, and cross the ball into the box. The strikers should be able to score goals"

- **Environment Dynamics Function:**

  "The behavior that is encouraged is when the forwards are restricted, the midfielder can support and take away the defenders. This is a very encouraging behavior because it allows the team to keep possession of the ball and control the game. You should only identify 5 types of rewards: 2: Optimal behavior; 1: Encouraging behavior; 0: Borderline behavior; -1: Punishing behavior; -2: Worest behavior."

- **Rewards Function:**

  "When the direction of shotting is vertical to the goal, the ball will be easy to the goal."

**{observation_space_desc}**

"First, it provides information such as the time and score of the match. Second, which side has control of the ball, and the active player in the Left team you need to propose corresponding policies to him.
Next is the position, direction and role information of each player:

- In this text description, the football grass field, which is 120m * 90m in the real world, is divided into 240 zones in the simulator, where "x" ranges from 1 to 20 (left to right from the left team's perspective) and "y" ranges from 1 to 12 (bottom to top).
- Based on this transformation, the concepts in real-world football field can be mapped to the simulator, For instance:
  - the center circle is at (10, 6), and the lower left corner from the left team's view is at (1, 1), and the upper right corner position of the Right team is (20, 12)
  - the distance from (10, 5) to (11, 5) in the real-world is 12m, and the distance from (10, 5) to (10, 6) is 7.5m
  - the left team's penalty area is zone (1, 4)-(1, 8)-(2, 8)-(2, 4), and the right team's penalty area is zone (20, 4)-(20, 8)-(18, 8)-(18, 4)
  - the center circle position of the field is zone (10, 6), where the game start
- The position of the player is defined by the player's moving in the corresponding direction, including north, northeast, east, southeast, south, southwest, west, northwest
- Role information: for each team, player roles are fixed, including Goalkeeper, Forward, Forward, Defender, Defender, Defender, Defender, Midfielder, Midfielder, Midfielders, Forward

NOTE: Venues never interchange or change. Following the position information is direction, meaning the direction the player is currently facing and the direction of future actions."

**{action_space_desc}** "The action set includes:

1. `action_idle`: A no-op action, sticky actions are not affected (player maintains his directional movement etc.)

2. `action_left`: Sticky action and will change the player's direction, player will continue to move left until another action is taken. Such as, from zone(11,4) to zone(10,4)

3. `action_top_left`: Sticky action and will change the player's direction, player will continue to move top left until another action is taken. Such as, from zone(11,4) to zone(10,5)

4. `action_top`: Sticky action and will change the player's direction, player will continue to move top until another action is taken. Such as, from zone(11,4) to zone(11,5)

5. `action_top_right`: Sticky action and will change the player's direction, player will continue to move top right until another action is taken. Such as, from zone(11,4) to zone(12,5)

6. `action_right`: Sticky action and will change the player's direction, player will continue to move right until another action is taken. Such as, from zone(11,4) to zone(12,4)

7. `action_bottom_right`: Sticky action and will change the player's direction, player will continue to move bottom right until another action is taken. Such as, from zone(11,4) to zone(12,3)

8. `action_bottom`: Sticky action and will change the player's direction, player will continue to move bottom until another action is taken. Such as, from zone(11,4) to zone(11,3)

9. `action_bottom_left`: Sticky action and will change the player's direction, player will continue to move bottom left until another action is taken. Such as, from zone(11,4) to zone(10,3)

10. `action_long_pass`: The player will long pass to their teammate in his current direction. Before you pass, you should release the sprint if the agent is doing sprint. A long pass covers a large distance on the field

11. `action_high_pass`: The player will high pass to their teammate in his current direction. Before you pass, you should release the sprint if the agent is doing sprint. A high pass sends the ball into the air, often over obstacles, to reach a teammate

12. `action_short_pass`: The player will short pass to their teammate in his current direction. Before you pass, you should tune make sure the direction is fine, using action 0-8 to change the direction. A short pass is a quick and close-range exchange between teammates, commonly used to maintain possession and build an attack

13. `action_shot`: Players will try to shoot. When near to the opponent penalty area, such as zone(x, y), x>18, 4<y<8, try to shoot

14. `action_sprint`: When the player will choose this action, the agent will sprint with sticky action's direction, it will make agent run faster

15. `action_release_direction`: Player will stop moving in the current direction. Choose it when you want this agent to change the direction, after this action, you should choose another action from 0-8 to change it

16. `action_release_sprint`: Player will stop sprinting, only when the agent's during the sprint

17. `action_sliding`: The player will try to slide the tackle. If your position is on the opponent's path with the ball, you can intercept it. However, if the sliding tackle fails, you will be separated by a large distance, allowing the opponent to ignore the defense

18. `action_dribble`: Players will try to dribble. When they have the ball, dribbling will greatly improve the success rate of dribbling, especially in multi-person double-teams and difficult-to-handle situations

19. `action_release_dribble`: Player will stop dribbling"

**{hint}**

- **Policy hint:**

    For example, if the active player is Player 2 and you want him to close to the ball and control it, in the texted observation:
    – Forward Player 2 is at Zone(9,9)
    – The ball is at Zone(11,8)

Your thought should be: "Given these coordinates, the ball is diagonally one zone to the right (east) and one zone down (south) from the player's current position. The most direct route to the ball would indeed be diagonally towards the bottom right. Therefore, the most appropriate action for Forward Player 2 in this situation would be: action_bottom_right"

- **Dynamics hint:**

  Important keys in the observation:
  - `score`: The current score of the match, should be a list of two integers, such as [0,0]
  - `step`: The current step of the match
  - `ball_ownership`: 1 for left team, 2 for right team, 0 for no team/during passing
  - `left_active_player_zone`: The zone of the active player [x,y]
  - `ball_zone`: Ball position [x,y]

  NOTE: if the active player still owner the ball, the player zone and the ball zone should be the same.

  Based on the current texted observation and the action set, generate the next step text observation.

  For instance, if the current observation shows the Left team player 3 in zone (12,4) and the chosen action is "Top," the next observation should show the player in zone (12,5).

- **Reward hint:**

  Based on the current texted observation and the action set, generate the dense reward of the current step's action and state, based on the reward code above. The rewards should be one of -2, -1, 0, 1, or 2.

**{inference_format_example}**

- **Policy Inference:**

```
{
    "thought": "give your thought to generate the action
    the current step's state, based on the policy code above.",
    "action": 0
}
```

- **Dynamics Inference:**

```
{
    "thought": "based on the current texted observation and the action set,
    the ball and player 3 are in the zone [9,4], and the active player
    is controlling the ball, and choose the action go top, thus consider
    the right have very low chance to intercept the ball, the ball
    will be in the zone [9,5] in the next time step.",
    "score": [0,0],
    "step": 2,
    "ball_ownership": 1,
    "ball_zone": [9,5],
    "left_active_player_zone": [9,5]
}
```

- **Reward Inference:**

```
{
    "thought": "give your thought to generate the dense
    reward of the current step's action and state,
    based on the reward code above.",
    "dense_reward": 1
}
```

## F.5 Introspecting based on the Imaginary Dataset

We implement CIQL based on the open source codes of CQL in d3rlpy [62], where the uncertainty of rewards and transitions are estimated by an ensemble of 20 Gaussian models based on 4-layer full-connected neural networks [6, 47], which are learned through standard log-likelihood maximization from the imaginary dataset.

## F.6 General Recipe for Prompt Design

The specific prompt design would significantly affect the quality of the LLM outputs and the algorithm built on them. Though URI does not focus on proposing new prompting techniques, we briefly summarize the general design principle as follows in our practice. These principles are adopted in implementations for both URI and all baseline methods in our experiments.

1. Let LLMs output the "thoughts/analyze" first before outputting the results. The concrete prompts following this principle can be found in the "Response example" part in our prompts in Appendix E;

2. Make the requirements as explicit as possible. The concrete prompts following this principle can be found in the "Requirements" part in our prompts in Appendix E;

3. Use pseudo-code instead of using natural language to represent the knowledge.

(1) and (2) are well-known principles and have been verified in other studies many times. For (3), we would like to highlight its effectiveness based on the following ablation study. We conduct the same experiment setting as Fig. 4(a) by using natural language to represent the knowledge. As shown in the table below, if we switch to the natural language representation, whatever the embedding models and retrieval techniques we used, the hit rate of the retrieval will drop a lot, as demonstrated by Tab. 4. We hope that this could provide insights for people interested in designing similar LLM-based frameworks for other decision-making tasks.

Table 4: Comparison of different knowledge representation

|                  | code  | natural language | perf drop rate (%) |
|------------------|-------|------------------|--------------------|
| embedding-baai   | 0.076 | 0.035            | 53.9               |
| embedding-openai | 0.085 | 0.075            | 11.8               |
| summary-baai     | 0.079 | 0.055            | 29.7               |
| summary-openai   | 0.077 | 0.056            | 27.3               |
| URI-baai         | 0.342 | 0.051            | 85.1               |
| URI-openai       | 0.338 | 0.056            | 83.4               |
| avg              | -     | -                | 48.5               |

## F.7 Hyper-parameters

Here, we list the important hyperparameters for URI here.

Table 5: URI Hyperparameters.

| Parameter | Value |
|-----------|-------|
| pieces of knowledge for each time of knowledge aggregation $N_{\text{agg}}$ | 4 |
| retrieved segments $N_{\text{ret}}$ | 15 |
| learning rate ($\lambda$) | 0.0001 |
| weight of transition $\eta_T$ | 0.5 |
| weight of reward penalties $\eta_R$ | 0.5 |
| horizon of rollout $H$ | 10 |
| weight of conservative loss $\alpha$ | 60 |
| number of ensemble $N_{\text{ens}}$ | 20 |

### F.8 URI Algorithm

This algorithm implements the URI method for policy learning from books. It consists of three main stages: **1) Understanding (L1-L2):** knowledge is extracted and aggregated from the input book using CodeExtractor and CodeAggregator. **2) Rehearsing (L3-L14):** An imaginary dataset is generated by iteratively applying knowledge retrieval, instantiation, and language model-based simulation. **3) Introspecting (L15-L16):** a policy is learned from the imaginary dataset using Conservative Imaginary Q-Learning (CIQL). The algorithm leverages language models $\mathbf{M}_\pi$, $\mathbf{M}_R$, and $\mathbf{M}_T$ for action selection, reward estimation, and state transition, respectively, bridging the gap between textual knowledge and reinforcement learning.

---

**Algorithm 4** URI (Understanding, Rehearsing, and Introspecting)

---

**Require:** Book $\mathcal{B}$, simulator information $|\mathcal{M}|$, LLM-injected dynamics function $\mathbf{M}_T$, reward function $\mathbf{M}_R$, and policy $\mathbf{M}_\pi$, imaginary dataset $\mathcal{D}_{\text{img}} \leftarrow \emptyset$.

1: $\mathcal{K}^0 \leftarrow$ CODE EXTRACTOR$(\mathcal{B})$        ▷ Sec. 5.2 (Algorithm 1)
2: $\mathcal{K}' \leftarrow$ CODE AGGREGATOR$(\mathcal{K}^0)$        ▷ Sec. 5.2 (Algorithm 2)
3: **for** $i = 1$ to num_trajectories **do**
4:     $\hat{s}_t \leftarrow$ SAMPLE INITIAL STATE$(|\mathcal{M}|)$
5:     **for** $t = 1$ to $H$ **do**        ▷ Sec. 5.3
6:        $\mathcal{K}^S \leftarrow$ KNOWLEDGE SCOPE RETRIEVAL$(\hat{s}_t, \mathcal{K}')$
7:        $K_\pi^I, K_R^I, K_T^I \leftarrow$ KNOWLEDGE INST$(\hat{s}_t, |\mathcal{M}|, \mathbf{R}_{\text{scope}}(\hat{s}_t, \mathcal{K}^S))$
8:        $\hat{a}_t \leftarrow \mathbf{M}_\pi(\hat{s}; K_\pi^I)$        ▷ Imaginary Action
9:        $\hat{r}_t \leftarrow \mathbf{M}_R(\hat{s}_t, \hat{a}_t, K_R^I)$        ▷ Imaginary Reward
10:      $\hat{s}_{t+1} \leftarrow \mathbf{M}_T(\hat{s}_t, \hat{a}_t, K_T^I)$        ▷ Imaginary Next State
11:      $\mathcal{D}_{\text{img}}.\text{add}(\hat{s}_t, \hat{a}_t, \hat{r}_t, \hat{s}_{t+1})$
12:      $\hat{s}_t \leftarrow \hat{s}_{t+1}$
13:     **end for**
14: **end for**
15: $\pi \leftarrow$ CIQL$(\mathcal{D}_{\text{img}})$        ▷ Sec. 5.4
16: **return** $\pi$

---

# G  Additional Ablations

## G.1  Data Aggregation Visualization for Tic-Tac-Toe

Similar to the visualization analysis in the main text, we also conduct t-SNE [59] visualization for the Tic-Tac-Toe environment to examine the quality of imaginary data generation and uncertainty estimation. As shown in Fig. 8, despite the discrete nature of the state space in Tic-Tac-Toe, we observe similar patterns as in the football environment. The imaginary data (separated into low and high uncertainty regions) largely follows the distribution of real data collected from optimal minimax gameplay. The uncertainty estimator successfully identifies deviant clusters (marked by yellow dashed circles) that lie outside the distribution of optimal play data. This consistency in behavior across both simple discrete environments and complex continuous ones supports the robustness of URI's data generation and uncertainty estimation mechanisms.

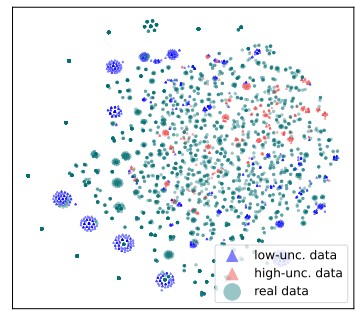

**Figure 8:** Visualization of the projected distributions for real and imaginary datasets in the Tic-Tac-Toe environment.

## G.2  Performance Comparison with Additional Baselines in GRF

To further validate the effectiveness of our approach, we implement three additional baselines: (1) RT-Policy: a policy trained using LLAMA following the RT-style architecture [63], using the same 7,500 initial states as URI and rule-based actions for training; (2) URI w/o BK: a variant of URI that generates imaginary data without using book knowledge; and (3) CQL: a pure offline RL approach using the Conservative Q-Learning algorithm with real interaction data.

**Table 6:** Performance Comparison of RT-policy, URI w/o book knowledge, CQL, Against Built-in AI Levels in a GRF 11 vs 11 settings, where the performance of URI is averaged among three different seeds, all methods are tested with 20 matches except RT-Policy which are tested with 10 matches due to computation resource limitation.

| Method | Easy | | | | Medium | | | | Hard | | | | Average |
|---|---|---|---|---|---|---|---|---|---|---|---|---|---|
| | W | D | L | GDM | W | D | L | GDM | W | D | L | GDM | GDM |
| *RT-Policy* | 0% | 50% | 50% | -2.10 | 0% | 20% | 80% | -1.60 | 0% | 10% | 90% | -3.20 | -2.29 |
| *URI w/o BK* | 30% | 55% | 15% | 0.16 | 22% | 58% | 20% | 0.04 | 22% | 50% | 28% | -0.06 | 0.04 |
| *CQL* | 33% | 49% | 18% | 0.20 | 36% | 49% | 15% | 0.22 | 17% | 53% | 30% | -0.20 | 0.07 |
| **URI (Ours)** | **37%±4%** | **57%±4%** | **6%±4%** | **0.40 ± 0.14** | **42%±12%** | **50%±8%** | **8%±4%** | **0.43 ± 0.24** | **32%±14%** | **58%±6%** | **10%±7%** | **0.32 ± 0.14** | **0.38 ± 0.05** |

# H  Examples of Data Generation and Policy Execution Videos

We list more visualization results for each stage of URI in the links: `https://plfb-football.github.io/`, including examples of knowledge extraction, code aggregation, imaginary data, and videos of policy execution in GRF.

# I  Open Problems and Future Directions

We hope that this promising result will initiate more research on PLfB. We will first elaborate on the points mentioned in Sec. 7 on applying URI when textual data are insufficient of low-quality.

- **Multimodal Data Integration**: Beyond purely textual data, additional modalities such as tutorial voices, demonstration videos, and replays can be incorporated. By employing advanced multimodal large language models, these diverse forms of data can be processed in a manner akin to the current handling within the URI, thereby augmenting the knowledge base and enhancing the robustness of policy learning.

- **Utilization of Real Interaction Data**: During the introspection phase, incorporating real interaction data with the target environment, rather than relying solely on simulated data generated by large language models, can enhance policy learning. This mixed-data approach can be used to further fine-tune various modules within the URI framework, potentially boosting overall performance. It also mimics human learning behavior better as humans learn from both reading and

actual experience. However, one might develop new techniques to utilize both sources of data for better policy learning.

- **Injection of Prior Knowledge**: The URI framework allows for the integration of human expert knowledge at different stages of the pipeline. Experts can provide specific code knowledge representations/formulations/templates when generating code-based knowledge. We can also provide constraints during the rehearsal process to enhance stability and realism.

- **Evaluating the Quality of Textual Data:** Most current studies in this fields [64–66] are still in primitives. These studies suggest frameworks and empirical strategies that can be adapted to evaluate the relevance and quality of textual and multimodal data for training purposes. However, given the significant modality gap between textual data and the neural network parameters in PLfB topics—and the largely unsupervised nature of our learning process—this is not a trivial problem. This complexity requires innovative approaches for assessment.

We are also compelled to investigate other potential applications of the framework beyond the scope of the current experiment. Domains such as embodied AI for robotics, which require a large amount of data for foundation policy learning and the robotics dataset currently, could benefit significantly from the integration of the paradigm for data augmentation. On the other hand, currently we just formulate the problem as an MDP and build a single-agent system for the football tasks, we believe that multi-agent systems can be built for this problem in the future work [67]. This exploration opens exciting avenues for future research and application of the method.

## J    Compute Resources

All experiments were conducted on a high-performance computing (HPC) system featuring 128 Intel Xeon processors running at 2.2 GHz, 5 TB of memory, an Nvidia A100 PCIE-40G GPU, and two Nvidia A30 GPUs. This computational setup ensures efficient processing and reliable performance throughout the experiments.

## K    Broader Impact Statement

Integrating Large Language Models (LLMs) with Reinforcement Learning (RL) in PLfB offers a groundbreaking approach to AI skill acquisition. This innovation has the potential to revolutionize industries reliant on automation, such as manufacturing and robotics, by enabling AI agents to learn complex tasks from textual sources, reducing the need for extensive real-world data. However, ethical considerations regarding bias, privacy, and transparency are paramount. Overall, PLfB promises enhanced efficiency and accessibility but requires careful navigation of its social implications.

