# OpenReview forum: "Policy Learning from Tutorial Books via Understanding, Rehearsing and Introspecting"
_NeurIPS.cc/2024/Conference — NeurIPS 2024 oral_

### Official Review · Reviewer_b6EF · 2024-07-12

**Soundness:** 3
**Presentation:** 2
**Contribution:** 3
**Rating:** 7
**Confidence:** 4

**Summary:**

The paper presents a method to distill knowledge about a given task or domain from text based knowledge into a form that can be used train a RL policy. The method extracts knowledge from text with a LLM, which is represented in a pseudocode-like textual form, and uses the knowledge to turn the LLM into a dynamics function, a reward function, and policy. Then the LLM is used to generate example trajectories by conditioning on the different types of extracted knowledge to turn it into a dynamics function, a reward function, and a policy. Finally conservative Q-learning is applied to the generated trajectories to learn a policy robust to the noise in the trajectory dataset. The method is evaluated on the Google Research Football environment against several baselines. The results demonstrate that the method improves performance relative to the baselines.

**Strengths:**

- Directly distilling knowledge from textual sources into control tasks is an important topic, and this paper takes a strong step in that direction.
- The paper's experiments are decently extensive and dig into the details about how/why URI has the observed improvement gains.

**Weaknesses:**

- It was difficult for me to follow what exactly what does in the method. The amount of notation is maybe obfuscating what the specifics of the method is.
- A class of methods that use LLMs as the backbone is missing, e.g. RT-2 or "Large Language Models as Generalizable Policies for Embodied Tasks"
- It would have been helpful to see how well URI does in a head-to-head match with the baselines (e.g. LLM-as-agent, LLM-RAG, and Random Policy). Given that the knowledge comes from books, which likely talk about semi-skilled strategies, it is unclear how important it is for the opponent to be of a similar quality as is discussed in the textual material. Head-to-head match ups would help understand how general and robust the policy is.

**Questions:**

- How does this method relate to the use of background textual knowledge in "Motif: Intrinsic Motivation from Artificial Intelligence Feedback" and "Learning to Model the World With Language"?
- How beneficial is learning with a LLM versus learning a policy with the task reward and no LLM?

**Limitations:**

The limitations are not discussed in the main body of the paper

---

> ### Author Rebuttal · Authors · 2024-08-07
>
> **Q1: It was difficult for me to follow what exactly what does in the method.**
>
> We are sorry that the current presentation of URI is not straightforward enough. As URI’s implementation involves interactions among several components from different research domains,  it indeed might look complex, especially for readers who are not familiar with one of these domains.  In response to this question, we plan to add a pseudo code to the revised paper (see Algorithm 3 in the attached PDF) to make the interactions unambiguous. Besides, we commit to open-source our full-procedure code to help users understand the details of this project. We kindly recommend the reviewer check our open-source plan in the global response.  We hope these efforts will make this study more easy to follow.
>
> **Q2: A class of methods that use LLMs as the backbone is missing.**
>
> Thanks for the valuable suggestion. RT is a classic baseline missing before. We implement an RT-style policy for football based on LLAMA and report the results in Table 2 and Figure 1 in the attached PDF. We use the same 7,500 initial states as URI and the rule-based actions to train the RT policy. After training, the loss can be reduced to normal as shown in Figure 1. However, though we tried our best to improve the performance, RT-policy turned out to fail in reaching the goal (See Table 2). We assume it is due to the size of training data is too small to be used for generalizable policy imitation. We will keep on trying but we cannot guarantee that we can get a better result.
>
> In response to this suggestion, we plan to (1) add RT-branch studies to the related work and (2) add the best performance we can reach to the revised paper.
>
> **Q3: It would have been helpful to see how well URI does in a head-to-head match with the baselines.**
>
> We agree that the head-to-head evaluation is valuable and will make the effectiveness of URI more convincing. Due to the limitation of the current simulator implementation, it will require considerable engineering work to modify the current Football environment to support the simultaneous evaluation of two policies. We are trying our best to finish such a job and commit to reporting the results in the revised version.
>
> On the other hand, the extended experiment in Tic-Tac-Toe does include head-to-head evaluations in the attached PDF. The results show that URI significantly outperforms the two major baselines LLM-as-agent and LLM-RAG with **+66%** and **+44%** net win rate. We hope this result can also solve the reviewer's concern.
>
> **Q4: How does this method relate to the use of background textual knowledge in "Motif: Intrinsic Motivation from Artificial Intelligence Feedback" and "Learning to Model the World With Language"?**
>
> Thanks the reviewer for pointing out these two related works. We will add a brief introduction and comparison with them in the related work section.  Specifically,
>
> 1. Motif: Intrinsic Motivation from Artificial Intelligence Feedback: the language is used as a caption of the observation. LLM is used as a surrogate reward model to output an evaluation (preference) of the observation (state) based on its caption. Such a preference model is then used to distill a reward model. Given this reward model, the policy is still trained in an online fashion. This could be classified into the LLM as a reward model category in our related work.
> 2. Learning to Model the World With Language: the observation contains both image and language. LLM is used as a part of the world model to predict future observations. The training paradigm is still the traditional Model-based RL, but the model here is multi-modal. This could be classified into the "LLM as dynamics model category" in our related work.
>
> **Q5: How beneficial is learning with an LLM versus learning a policy with the task reward and no LLM?**
>
>  Since the PLfB is more like in a offlineRL setting, we tested the performance of CQL as the result of policy learning with the task reward and no LLM. The results are in Table 2 in the attached PDF. In summary, CQL also achieved a competitive performance compared with other baselines, i.e., LLM-as-agent and LLM-RAG, but still significantly underperforms URI (0.07 vs 0.38 in average GDM). This result demonstrates the benefits of URI in improving the policy's performance compared with standard policy learning methods.
>
> **Q6: The limitations are not discussed in the main body of the paper.**
>
> We would like to kindly point out that the limitation of this paper is mentioned in Section 7. Since the space is limited in the main body, we leave a link to Appendix F in Section 7, where give our full discussions and limitations of this study.

---

> > ### Comment · Reviewer_b6EF · 2024-08-09
> >
> > The authors have addressed my primary concern by committing to provide pseudocode and make their code open source. Please also find space to move a discussion of the limitations to the main body, and I will raise my score.

---

> > > ### Author Response · Authors · 2024-08-09
> > >
> > > Thank you for your positive feedback on our work. We are pleased to have addressed your primary concern. We commit to put the discussion of the limitations to the main body of the paper in the revision, utilizing the extended page limit. Thank you again for your suggestions!

---

### Official Review · Reviewer_mSmU · 2024-07-14

**Soundness:** 3
**Presentation:** 3
**Contribution:** 3
**Rating:** 7
**Confidence:** 4

**Summary:**

The paper introduces a novel approach to policy learning, termed "Policy Learning from Books," which leverages existing textual knowledge, such as books and tutorials, to develop policy networks without the need for extensive real-world interactions. This method is inspired by how humans learn new skills from written resources. The authors propose a three-stage framework called URI (Understanding, Rehearsing, and Introspecting) to implement this approach. In the URI framework, the process begins with understanding the content of the books, followed by rehearsing decision-making scenarios based on the understood knowledge, and finally, introspecting on these rehearsed scenarios to refine a policy network. To demonstrate the effectiveness of this method, the researchers applied it to train a football-playing policy using the Google Football game. The trained agent, which did not interact with the environment during training, achieved a 37% winning rate against the built-in AI, significantly outperforming a GPT-based agent that only managed a 6% winning rate. This study highlights the potential of utilizing textual knowledge for enhancing decision-making processes in reinforcement learning.

**Strengths:**

1. I think the studied topic: policy learning from book is interesting and meaningful to the community. Maybe it can be adopted as a kind of new policy learning paradigm from the novel data sources, beyond the traditional reinforcement learning and imitation learning. It somehow provides the potential that does not require the extensive agent-enviroment interaction data anymore.
2. From the paper, the proposed method framework including the understanding, rehearsing, and introspecting is reasonable and intuitive.
3. The writing of this paper is easy to follow and the paper structure has been carefully organized.
4. The empirical results present in the experiment part are generally persuasive.

**Weaknesses:**

1. More datasets can be considered in the experiments.
2. I think the potential application of this work is beyond the football game. Why not validate its effectiveness on learning some other policies and validating their performance in some different challenging environments?
3. As for the baselines, the authors only compare their proposed framework with the LLM-based and rule-based policies. Why not compare it with some other policies learned with conventional RL algorithms, like PPO, DDPG, SAC, and so on?

**Questions:**

1. I am curious to the performance of distilled policy from the imaginary dataset generated directly by the GPT without the information from the books?
2. Why not provide the pseucode for this work in the anonymous repository? Maybe the reproducibility of this work can be further enhanced.
3. The caption and illustration of Figure 1 are a little bit confusing. Personally, I think book tutorial is also a kind of data, though it is not the real interaction data between the agent and the interaction. I suggest the authors revise this point in the future version.
4. The Figure 2 is somewhat redundant, considering the detailed 3-stage framework: understanding, rehearsing, and introspecting, has been clearly provided in the Figure 3.

**Limitations:**

See above weaknesses and questions.

---

> ### Author Rebuttal · Authors · 2024-08-07
>
> **Q1: add baselines: (1) distilled policy from the imaginary dataset generated directly by the GPT without the information from the books; (2) compare it with some other policies learned with conventional RL algorithms**
>
> Thanks for the valuable suggestions.  In the rebuttal period, we implement the two baselines named “URI w/o BK” and “CQL”, which can be seen in Table 2 in the attached file. Since the PLfB is more like an offline setting, we use a popular offline RL baseline CQL to demonstrate the results of conventional RL algorithms. We found that pure URI without book knowledge and pure offline RL algorithm can reach competitive performance compared with the baselines but URI still achieves significantly better results (0.38 vs 0.04 in average GDM for “URI w/o BK” and 0.38 vs 0.07 for “CQL”). The results demonstrate that purely using the dataset or the inner knowledge of LLMs cannot should the problems. We kindly recommend the reviewer check the detailed results in the attached PDF.
>
> In response to the suggestion, we commit to adding these two baselines to our revised paper. We believe it will make the effectiveness of URI more convincing.
>
> **Q2: Why not provide the pseucode for this work in the anonymous repository? Maybe the reproducibility of this work can be further enhanced.**
>
> We plan to add a pseudo code to the revised paper (see Algorithm 3 in the attached PDF) to make the interactions among these components unambiguous. Besides, we commit to open-source high-quality code to help users understand the full details of this project. The global response contains the complete plan and scope of the open-sourcing.
>
> **Q3: The caption and illustration of Figure 1 are a little bit confusing. Personally, I think book tutorial is also a kind of data & The Figure 2 is somewhat redundant, considering the detailed 3-stage framework: understanding, rehearsing, and introspecting, has been clearly provided in the Figure 3.**
>
> Thanks to the reviewer for pointing out these two presentation problems. We agree that the book tutorial can also be regarded as a source of data. We will change it to “interaction trajectories” in the revised version. Figure 2 is the proposed general URI methodology for the problem of PLfB and we expect the readers to quickly develop an idea about our methodology. Figure 3 is more about solution implementation: it provides an overview of the exact implementation for PLfB in this study. We will improve both figures and captions to make such a distinction more clear.
>
> **Q4: More datasets & applications can be test.**
>
> We acknowledge that applying URI in just single domains is not enough to demonstrate the generalizability of the methodology. To alleviate such a concern, we build a new proof-of-concept benchmark based on the classic Tie Tac Toe game (TTT) and verify URI's performance on it. In short, URI continues to behave well. The benchmark setting and more results are elaborated in the global response

---

> > ### Comment · Reviewer_mSmU · 2024-08-09
> > **Response to Author's Rebuttal**
> >
> > Thanks for your detailed response. Most of my concerns have been addressed. I will raise the score.

---

> > > ### Author Response · Authors · 2024-08-09
> > >
> > > Thank you for your valuable feedback and for adjusting your evaluation of our work. We appreciate your acknowledgment of our efforts. We commit to adding the mentioned experiments to our revised paper.

---

### Official Review · Reviewer_HEXR · 2024-07-15

**Soundness:** 3
**Presentation:** 2
**Contribution:** 2
**Rating:** 6
**Confidence:** 4

**Summary:**

The paper introduces an intriguing approach to reinforcement learning (RL) through the concept of Policy Learning from Books (PLfB), which leverages textual resources like books and tutorials to derive policy networks. This methodology represents a interesting departure from traditional RL techniques that rely heavily on real interactions with the environment as recall by the authors.

The proposed URI framework outlines how the system first comprehends the textual content, then rehearses decision-making trajectories, and finally introspects to refine the policy network using an imaginary dataset.

The practical validation of this method is demonstrated by training a football-playing policy and testing it in the Google Football simulation environment. The results are promising, with the agent achieving a 37% winning rate against the built-in agent without any interaction with the environment during training. This is a substantial improvement over the 6% winning rate achieved using an LLM.

In addition, the paper addresses the question of extracting policies without direct environment interaction by incorporating descriptions of MDP structures, transition functions, and reward functions within the textual data. This ensures the feasibility of the PLfB approach and adds depth to the methodology.

However, one aspect that could have been elaborated on is the influence of the prompting strategy used for generating the imaginary dataset. Detailing how different prompting techniques impact the quality and effectiveness of the dataset could provide valuable insights and enhance the robustness of the proposed approach.

Overall, the paper is a interesting contribution to the field, proposing an original perspective on utilizing textual knowledge for policy learning wel align with this current epoch of LLMs development. The results are encouraging, and the methodology is articulated and validated through practical experiments.

**Strengths:**

* Innovative Concept: Introduces the novel idea of Policy Learning from Books (PLfB), leveraging textual resources for policy network derivation, which is a significant departure from traditional RL methods.
* Human-Like Learning Process: The URI framework—understanding, rehearsing, and introspecting—mimics how humans learn from books, making the approach intuitive and biologically inspired.
* Promising Results: Demonstrates a 37% winning rate in the Google Football simulation environment, significantly outperforming a Large Language Model (LLM) which achieved only a 6% winning rate.
* Feasibility and Depth: Incorporates detailed descriptions of MDP structures, transition functions, and reward functions within the textual data, ensuring the feasibility of extracting useful policies without direct environment interaction.
* Practical Validation: The methodology is well-validated through practical experiments, strengthening the credibility and significance of the research.
* Alignment with Current Trends: The approach aligns well with the current advancements in LLMs, making it relevant and timely.

**Weaknesses:**

* Prompting Strategy Details: The paper lacks detailed discussion on the influence of the prompting strategy used for generating the imaginary dataset. Exploring different prompting techniques could provide valuable insights and improve the approach's robustness.
* Textual Resource Dependence: The success of the approach heavily relies on the quality and comprehensiveness of the textual resources, which might limit its applicability in domains with sparse or low-quality textual data.
* Generalizability: The generalizability of the method across different domains remains uncertain and needs further exploration to ensure its broad applicability.
* Complexity of Implementation: The methodology, while innovative, might be complex to implement and require significant computational resources, which could be a barrier for some researchers or practitioners.

**Questions:**

1. Prompting Strategy Exploration: Could you elaborate on the specific prompting strategies used to generate the imaginary dataset? How do you believe different strategies might influence the quality and effectiveness of the policy learned?

2. Generalizability Across Domains: What steps do you envision for testing the generalizability of the Policy Learning from Books (PLfB) approach in different domains or environments? Have you considered any preliminary experiments in varied contexts?

3. Quality of Textual Resources: How do you plan to address potential limitations related to the quality and comprehensiveness of the textual resources used? Are there specific criteria or methods you would recommend for selecting or evaluating these resources?

---

> ### Author Rebuttal · Authors · 2024-08-07
>
> **Q1: Prompting Strategy Details & Exploration: Could you elaborate on the specific prompting strategies used to generate the imaginary dataset? How do you believe different strategies might influence the quality and effectiveness of the policy learned?**
>
> We agree that the design of prompts matters in LLM-related methods. Though we do not focus on propose new prompting techniques,  in our practice, the following principles of prompt designs are important in this projects:
>
> 1. Before output the results, let LLMs output the “thoughts/analyze” firstly, and we will give an example of thoughts in the prompts. Reference to the “Response example“ parts in our prompts in Appendix C and D for more details.
> 2. Make our requirements as explicit as possible. Reference to the “Requirements“ parts in our prompts in Appendix D.
> 3. As mentioned in Section 5.2, instead of using natural language to represent the knowledge, we use pseudo-code.
>
> Since (1) and (2) are well-known principles and have been verified in other studies many times, in the following, we ablate the effects of knowledge representation. We conduct the same experiment setting as Figure 5(a) in the main body by using natural language to represent the knowledge. As shown in the table below, if we switch to the natural-language-representation, whatever the embedding models and retrieval techniques we used,  the hit-rate of the retrieval will drop a lot.
>
> |  | code | natural language | perf drop rate (%) |
> | --- | --- | --- | --- |
> | embedding-baai | 0.076 | 0.035 | 53.9% |
> | embedding-openai | 0.085 | 0.075 | 11.8% |
> | summary-baai | 0.079 | 0.055 | 29.7% |
> | summary-openai | 0.077 | 0.056 | 27.3% |
> | URI-baai | 0.342 | 0.051 | 85.1% |
> | URI-openai | 0.338 | 0.056 | 83.4% |
> | avg | / | / | 48.5% |
>
> In response to this concern, we commit to: (1) describe the key prompting strategies we used to implement URI to the revised paper; (2) add the above results to the revised paper.
>
> **Q2: Generalizability Across Domains.**
>
> Thanks for the valuable question. We acknowledge applying URI in just single domains might raise concerns about the generalizability of the methodology. In response to this concern, we build a new proof-of-concept benchmark based on the classic Tie Tac Toe game (TTT) and verify URI. We kindly recommend the review to check the details of the results in the first section of the global response letter.
>
> **Q3: Complexity of Implementation**
>
> We acknowledge the reviewer's concern on the complexity of the implementation. In response to the concern, we would like to use a high-quality open-source code to keep this study easy to follow and reproduce. We kindly recommend the review to check  our open-source plan in the second section of the global response letter.
>
> **Q4: Quality of Textual Resources**
>
> *Q4.1: How do you plan to address potential limitations*
>
> The effectiveness of the policy derived by the URI framework fundamentally hinges on the quality of the textual data. It is indeed important that the textual resources should cover sufficiently the dynamics, policy, and rewards of the targeted environment so that relevant knowledge and imaginary data can be extracted to train the policy, otherwise it is impossible to derive a good enough policy.  There are several potential approaches to address this limitation:
>
> 1. **Multimodal Data Integration**: Beyond purely textual data, additional modalities such as tutorial voices, demonstration videos, and replays can be incorporated. By employing advanced multimodal large language models, these diverse forms of data can be processed in a manner akin to the current handling within the URI, thereby augmenting the knowledge base and enhancing the robustness of policy learning.
> 2. **Utilization of Real Interaction Data**: During the introspection phase, incorporating real interaction data with the target environment, rather than relying solely on simulated data generated by large language models, can enhance policy learning. This mixed-data approach can be used to further fine-tune various modules within the URI framework, potentially boosting overall performance. However, one might develop new techniques to utilize both sources of data for better policy learning.
> 3. **Injection of Prior Knowledge**: The URI framework allows for the integration of human expert knowledge at different stages of the pipeline. Experts can provide specific code knowledge representations/formulations/templates when generate code-based knowledge.  We can also provide constraints during the rehearsal process to enhance stability and realism.
>
> *Q4.2: Criteria for Selecting and Evaluating Resources*
>
> Evaluating the quality of training data remains a challenging issue across machine learning community. There are possible methods [1-3] that might guide our evaluation. These studies suggest frameworks and empirical strategies that can be adapted to evaluate the relevance and quality of textual and multimodal data for training purposes. However, given the significant modality gap between textual data and the neural network parameters in PLfB topics—and the largely unsupervised nature of our learning process—this is not a trivial problem. This complexity requires innovative approaches for assessment.
>
> [1] QuRating: Selecting High-Quality Data for Training Language Models
>
> [2] An Empirical Exploration in Quality Filtering of Text Data
>
> [3] DoGE: Domain Reweighting with Generalization Estimation
>
> We believe these discussions are valuable for the research community. In response to this question, we commit to include them in Appendix F of the revised paper. Additionally, we would like to highlight that many real-world decision-making scenarios have an abundance of textual tutorial resources, such as medical diagnosis, financial trading, software development, and educational tutoring. Thus, even in domains with rich textual resources, the topic remains broad and highly valuable.

---

> > ### Comment · Reviewer_HEXR · 2024-08-12
> >
> > I do appreciate the authors' review. Considering other reviews, I keep my score unchanged also.

---

> > > ### Author Response · Authors · 2024-08-13
> > >
> > > Thank you for your constructive feedback and for maintaining your score after considering the reviews. We appreciate your recognition and will continue to address any aspects highlighted throughout the review process to improve our work further.

---

### Official Review · Reviewer_uFxN · 2024-07-22

**Soundness:** 3
**Presentation:** 3
**Contribution:** 4
**Rating:** 8
**Confidence:** 3

**Summary:**

This paper introduces Policy Learning from Books (PLfB). This framework leverages the knowledge encoded in textual books and tutorials to train decision-making policies, specifically for playing football, without requiring direct interaction with the environment. This method is a three-stage framework Understanding, Rehearsing, and Introspecting -- **Unstanding**: extracts knowledge from books, uses it to **rehearse** decision-making trajectories in an imaginary dataset, and then **introspects** on the imagined data to distill a refined policy network.

They found that the URI approach significantly outperforms baseline methods in the Google Research Football (GRF) 11 vs 11 scenarios, and that the iterative process of code extraction and aggregation significantly reduces the number of code segments for dynamics, policy, and reward functions.

**Strengths:**

1. This paper is novel in terms of the method proposed. The URI framework is very intuitive and could be applied to other domains of agent learning.
2. Using imaginary dataset generated based on extracted knowledge is an interesting idea for synthetic data generation.
3. The efficiency of the method is also impressive for real world applications.

**Weaknesses:**

1. The main weakness could the narrow scope of application in this paper. It is not sure how data quality from the collected textbook data could affect the performance the model. If there are more domains where the authors could experiment with different data sources, the readers would have better expectation of the model
2. The second weakness is the complex framework has many components. This on one hand is the novelty of this paper, however, it also adds to the difficulty of replicating this method, especially on other domains.

**Questions:**

Can the URI framework be iterated -- collecting data online and summarize new knowledge. And the knowledge can be passed on to the next iteration of URI.

**Limitations:**

Yes.

---

> ### Author Rebuttal · Authors · 2024-08-07
>
> **Q1: If there are more domains where the authors could experiment with different data sources, the readers would have better expectation of the model.**
>
> We agree that applying URI to more domains would strengthen readers' understanding, expectation, and belief in it. Thus, we build a new proof-of-concept benchmark based on the classic Tie Tac Toe game (TTT) and verify URI. We kindly recommend the review to check the details of the results in the first section of the global response letter.
>
> **Q2: Concern about the complex framework, which has many components. This on one hand is the novelty of this paper, however, it also adds to the difficulty of replicating this method, especially in other domains.**
>
> We acknowledge the reviewer's concern about the complexity of the implementation. In response to the concern, we would like to use high-quality open-source code to keep this study easy to follow and reproduce. We kindly recommend the review to refer to the second section of the global response letter for our open-source plan. We will also continue to improve the text to make the whole paper more accessible.
>
> **Q3: Can the URI framework be iterated -- collecting data online and summarize new knowledge. And the knowledge can be passed on to the next iteration of URI.**
>
> It is an open question but we truly believe the future of URI should be iterative. It is intuitive since even when humans learn a new skill from books, it is not enough to be competent in tasks by just learning from books. We need real-world practices to bridge the knowledge-utilizing gaps in books. However, it is non-trivial to design a practical method to fully utilize the **online data to improve the URI pipeline.** We have made a brief discussion about such an extension as a future research direction in Appendix F. We are excited to dive deeper into this as future work.

---

> > ### Comment · Reviewer_uFxN · 2024-08-12
> >
> > I appreciate the authors' review. After reading the other reviews, I will keep my positive score unchanged.

---

> > > ### Author Response · Authors · 2024-08-13
> > >
> > > Thank you very much for your strong support and positive feedback on our manuscript. We are grateful for your recognition of the significance and contributions of our work. We will continue to address the key points discussed during the review process to further enhance the quality and impact of our research.

---

### Author Rebuttal · Authors · 2024-08-07

We thank all the reviewers for their constructive and thoughtful feedback. We appreciate all the recognition and kind comments on our work, including **conceptual novelty and enlightenment** (R1, R2, R3, R4); **realistic, extensive, and well-motivated experiments** (R1, R2, R3, R4); **promising results** (R1, R2, R3);  **intuitive motivation** behind the method design (R1,R2,R3). Beyond the work itself, reviewers also recognize that ideas in our work could be further utilized under a **broader context** including synthetic data generation and learning from richer resources (R1, R3).

In the following, we report our responses to the common concerns and suggestions proposed by the reviewers.

### **1 Generalizability of the URI methodology to other domains**

We acknowledge applying URI in just single domains might raise concerns about the generalizability of the methodology. In response to this concern, we build a new proof-of-concept benchmark based on the classic Tie Tac Toe game (TTT). In particular, we use a minimax policy, which is the optimal policy in this game, to collect all possible trajectories in this game. Then, for each trajectory, we use GPT to derive textual books by summarizing the trajectories, and analyzing the game mechanics, winning conditions, and strategic principles we can learn from the trajectory. Since we have all optimal trajectories, the process guarantees the textual book is high-quality to cover complete knowledge in this game (ignore the information loss of GPT generations) so it is an ideal testbed for PLfB.

We then apply URI in the TTT textual book. *Note that we use the same prompt template and just modify the task-specific contents. Our results can then be achieved by tuning the weight of transition, reward penalty, and conservative loss in CIQL, which are 0.05, 0.05, and 0.1 respectively*. The results are in Table 1, and Figure 2 in the attached PDF file. Our key findings are as follows:

1. As shown in Table 1, URI demonstrates superior performance across all opponents in head-to-head matches, where it achieves the highest net win rate (win - loss) of **+66%, +44%, +52%** against LLM-as-agent, LLM-RAG, and Random Policy, respectively.
2. Besides, against Minimax-noise, URI **still maintains a positive win-loss percentage**, indicating competitive ability even when facing a near-to-optimal strategy, while all baselines can only get negative net win rates.
3. We also apply several important experiments to show the effectiveness of URI components in the main body to the TTT environment. In particular, knowledge segment aggregation (Figure 4 in the main body) is in Figure 2 and visualization of the projected distributions for real and imaginary datasets (Figure 7 in the main body) is in Figure 3. The results are similar among these two environments, which further demonstrate the effectiveness of URI components. In particular, for knowledge segment aggregation, **the number of knowledge pieces will be reduced and converge after 4 iterations**; for visualization, **high-uncertainty regions can be identified** by the uncertainty predictor (marked with yellow ovals), while low-uncertainty regions (marked with blue ovals) are generated surrounded by the real data. We also found that the generated data cannot cover the real optimal trajectories, which indicates that there is still room to improve the quality of the trajectory imaginary in the rehearsing stage.


We again sincerely thank the reviewers for this valuable question. We believe that these results strongly demonstrate the generalizability of the URI methodology to other domains and further increase the quality and potential impacts of this study. We also commit to open-source the code and data of this additional experiment, such that researchers can refer to the modifications made in this experiment and adapt URI to other domains that they are interested in.

### **2 Complex to implement URI, especially in other domains**

We acknowledge the reviewer's concern about the complexity of the implementation. In response to the concern, we would like to use high-quality open-source code to keep this study easy to follow and reproduce. In particular, we commit to:

1. *Open-source the two environments*: there are several designs based on the original simulator to run our experiments, especially the transformations between the text-based and vector-based state/action space (as shown in Appendix A). We will open-source our Football environment and Tie Tac Toc environments with full details in our experiments to help researchers with further development.
2. *Open-source the full URI procedure*:  all the data collecting and training scripts of the full URI procedure, with the generated data in the process, including the results of Football tasks and TTT tasks will be open source.
3. *Configurable implementation*: To meet the requirements of users who want to quickly check the details of URI or test URI on other domains, we will refactor our code and use a single configuration file to all set/get the domain-specific config/information.

---

### Author Response · Authors · 2024-08-13

Dear Reviewers and ACs,

As the discussion period is almost over, we would like to deeply thank all reviewers and ACs for their efforts in evaluating the paper and for the timely responses during the discussion periods. We also appreciate all reviewers’ positive evaluations and recognition of our work's conceptual novelty, its value to the community, well-motivated methodology, extensive experiments, and promising results.

During the rebuttal periods, we received many valuable suggestions for improving the presentation of the paper, making it easier to follow and reproduce, and enhancing its potential impact. We summarize the revision plan we committed to as follows:

1. **Presentation:**
    - Add pseudocode to clarify the interactions of the components (R3, R4).
    - Polish Figures 1 and 2 (R3).
    - Include related work on Motif and Dynalang (R4).
    - Discuss the potential limitations when the quality of textual resources is low (R2).
    - Discuss the limitations of the paper in the main body instead of in the appendix (R4).
2. **Experiment:**
    - Add the results of the proof-of-concept benchmark based on the classic Tic Tac Toe game (TTT) to the paper (R1, R2, R3).
    - Add baselines, including URI-without-book-knowledge, pure RL-based algorithm, and RT-based method (R3, R4).
    - Include head-to-head match results between URI and the baselines (R4).
3. **Reproducibility:**
    - Open-source high-quality code, including the two environments, the full URI procedure, and a configurable implementation (R1, R2, R3, R4).
    - Discuss the details of the prompting strategy and highlight the important principles that make the solution work in the paper (R2).

Thank you again for your constructive feedback and recognition of our work. We are committed to revising the paper as planned to ensure it is well-represented and meets the reviewers’ expectations.

Sincerely,

Authors

R1: uFxN, R2: HEXR, R3: mSmU, R4: b6EF

---

### Decision · Program_Chairs · 2024-09-25

**Decision:**

Accept (oral)

**Comment:**

The paper introduces a novel and timely approach to solve reinforcement learning problems by incorporating textual knowledge about the problems. The experiments convincingly show that the approach can be effective, and significantly better than RL baselines and ablations of the approach.

All of the reviewers agree that the paper is technically sound, clearly written, with high significance and novelty. The authors in their feedback provided additional experiments and discussion that will make the revised paper stronger.